# Mechanism of pharmacochaperoning in a mammalian K<sub>ATP</sub> channel revealed by cryo-EM

# Mechanism of pharmacochaperoning in a mammalian $K_{ATP}$ channel revealed by cryo-EM

Gregory M Martin[1†], Min Woo Sung[1†], Zhongying Yang[1], Laura M Innes[1], Balamurugan Kandasamy[1], Larry L David[1], Craig Yoshioka[2]*, Show-Ling Shyng[1]*

[1]Department of Biochemistry and Molecular Biology, Oregon Health & Science University, Portland, United States; [2]Department of Biomedical Engineering, Oregon Health & Science University, Portland, United States

**Abstract** ATP-sensitive potassium ($K_{ATP}$) channels composed of a pore-forming Kir6.2 potassium channel and a regulatory ABC transporter sulfonylurea receptor 1 (SUR1) regulate insulin secretion in pancreatic β-cells to maintain glucose homeostasis. Mutations that impair channel folding or assembly prevent cell surface expression and cause congenital hyperinsulinism. Structurally diverse $K_{ATP}$ inhibitors are known to act as pharmacochaperones to correct mutant channel expression, but the mechanism is unknown. Here, we compare cryoEM structures of a mammalian $K_{ATP}$ channel bound to pharmacochaperones glibenclamide, repaglinide, and carbamazepine. We found all three drugs bind within a common pocket in SUR1. Further, we found the N-terminus of Kir6.2 inserted within the central cavity of the SUR1 ABC core, adjacent the drug binding pocket. The findings reveal a common mechanism by which diverse compounds stabilize the Kir6.2 N-terminus within SUR1's ABC core, allowing it to act as a firm 'handle' for the assembly of metastable mutant SUR1-Kir6.2 complexes.
DOI: https://doi.org/10.7554/eLife.46417.001

*For correspondence:
yoshiokc@ohsu.edu (CY);
shyngs@ohsu.edu (S-LS)

[†]These authors contributed equally to this work

**Competing interests:** The authors declare that no competing interests exist.

## Introduction

ATP-binding cassette transporters (ABC transporters) comprise a large protein superfamily responsible for transporting diverse molecules across cell membranes (*Thomas and Tampé, 2018*; *Trowitzsch and Tampé, 2018*; *Wilkens, 2015*). Uniquely, the sulfonylurea receptors SUR1 and SUR2 lack transport activity per se but instead are devoted to forming the ATP-sensitive potassium ($K_{ATP}$) channels, wherein four SUR.x subunits form a complex with four subunits of an inwardly rectifying potassium channel, Kir6.1 or Kir6.2 (*Aittoniemi et al., 2009*; *Ashcroft and Ashcroft, 1990*; *Bryan et al., 2007*; *Nichols, 2006*). $K_{ATP}$ channels are gated by intracellular ATP and ADP, key features which enable them to regulate membrane excitability according to the energetic state of the cell (*Nichols, 2006*). Expressed in many cell types, they have critical roles in endocrine, cardiovascular, neurological and muscular functions (*Aguilar-Bryan and Bryan, 1999*; *Foster and Coetzee, 2016*; *Nichols et al., 1996*). In pancreatic β-cells, $K_{ATP}$ channels composed of SUR1 and Kir6.2 couple glucose metabolism to insulin secretion (*Aguilar-Bryan et al., 2001*; *Ashcroft, 2005*). Gain-of-function mutations in the SUR1 gene *ABCC8* or the Kir6.2 gene *KCNJ11* are a major cause of neonatal diabetes, while loss-of-function mutations result in congenital hyperinsulinism (*Ashcroft et al., 2017*; *Koster et al., 2005*; *Stanley, 2016*). Among the latter, severe hyperinsulinism requiring pancreatectomy to prevent life-threatening hypoglycemia is often the result of $K_{ATP}$ channel mutations that reduce $K_{ATP}$ channel density on the β-cell surface by impairing channel biogenesis, assembly, and trafficking (referred to as trafficking mutations hereinafter) (*Huopio et al., 2002*; *Vajravelu and De León, 2018*).

Previously, we showed that $K_{ATP}$ channel inhibitors commonly used to treat Type 2 diabetes are efficient pharmacochaperones for correcting surface expression defects of certain trafficking-impaired mutant $K_{ATP}$ channels, including sulfonylureas (SUs) and glinides (*Yan et al., 2004*; *Yan et al., 2007*). Most trafficking mutations identified to date are in SUR1, perhaps due to its large size compared to Kir6.2 (*Martin et al., 2013*; *Snider et al., 2013*). The SUR1 protein has an N-terminal transmembrane domain referred to as TMD0 that interacts directly with the Kir6.2 subunit (*Babenko and Bryan, 2003*; *Chan et al., 2003*; *Lee et al., 2017*; *Li et al., 2017*; *Martin et al., 2017b*; *Schwappach et al., 2000*), followed by a cytoplasmic loop termed L0, and then the ABC core structure comprising two transmembrane domains TMD1 and TMD2 and two nucleotide binding domains NBD1 and NBD2 (*Aguilar-Bryan et al., 1995*; *Martin et al., 2017b*). Of particular interest, while trafficking mutations are found throughout SUR1, SUs and glinides have to date only corrected defects caused by mutations located in TMD0, the domain contacting Kir6.2. Recently, we found that carbamazepine (CBZ), an anticonvulsant structurally not related to SUs and glinides, also corrects $K_{ATP}$ channel trafficking defects (*Chen et al., 2013*; *Sampson et al., 2013*) and surprisingly is effective only for mutations in TMD0 of SUR1 (*Devaraneni et al., 2015*; *Martin et al., 2016*). Furthermore, like SUs and glinides, CBZ inhibits $K_{ATP}$ channel activity by reducing channel $P_o$ and abolishing channel responsiveness to MgADP (*Zhou et al., 2014*). The striking similarities of SUs, glinides and CBZ in their effects on the channel despite their chemical uniqueness suggest a shared pharmacochaperoning mechanism which remains elusive.

In this study, we took a comparative structural approach to understand how glibenclamide (GBC, a sulfonylurea), repaglinide (RPG, a glinide), and CBZ, interact with the channel to affect channel biogenesis and function. Using single-particle cryo-electron microscopy (cryoEM), we determined structures of the channel in the presence of GBC, RPG, and CBZ, and without pharmacochaperone. The structures show that like GBC (*Martin et al., 2017a*; *Wu et al., 2018*), RPG and CBZ occupy the same binding pocket in the transmembrane bundle above NBD1 of SUR1. Further, we undertook structural, biochemical and functional studies to determine the involvement of the distal N-terminal 30 amino acid stretch of Kir6.2, which is functionally critical to the actions of these drugs but whose structure has not yet been clearly defined. The combined results from these studies provide strong evidence that the distal Kir6.2 N-terminus is located in the cavity formed by the two halves of the SUR1 ABC core and is adjacent to the drug binding pocket. The study reveals how a chemically diverse set of $K_{ATP}$ channel inhibitors allosterically control channel gating, and also promote the assembly and trafficking of nascent channels to the cell surface, by stabilizing a key labile regulatory interaction between the N-terminus of Kir6.2 and the central cavity of the ABC core of SUR1.

## Results

### Structure determination

For structure determination, a FLAG-tagged hamster SUR1 and a rat Kir6.2 (95% and 96% identical to human, respectively) were overexpressed in the insulinoma cell line INS-1, and the channel complex affinity purified via the FLAG-epitope tag as described for our recent cryoEM structure determination of $K_{ATP}$ bound to GBC and ATP (*Martin et al., 2017a*; *Martin et al., 2017b*). These channels have been used extensively for structure-function and pharmacochaperone studies and are thus well characterized (*Inagaki et al., 1995*; *Shyng et al., 1998*; *Yan et al., 2004*). To allow direct comparison of channel structures containing bound RPG or CBZ with the GBC/ATP-bound structure solved previously (*Martin et al., 2017a*), data were similarly collected in the presence of ATP (1 mM without $Mg^{2+}$) but varying the pharmacochaperone by alternatively including 30 μM RPG (referred to as the RPG/ATP state), 10 μM CBZ (referred to as the CBZ/ATP state) or the drug vehicle 0.1% DMSO (referred to as the ATP-only state) in the sample. Further, as an additional control we determined the structure of channels without any pharmacochaperone or ATP (referred to as the apo state).

Initial data processing in RELION with C4 symmetry imposed yielded one dominant 3D class for each dataset, with an overall reported resolution ranging from ~4 Å for the RPG/ATP and CBZ/ATP states to ~7 Å for the ATP-only and the apo states (*Figure 1*; *Figure 1—figure supplements 1–4*; *Table 1*). As with the GBC/ATP state structures we reported previously (*Martin et al., 2017a*; *Martin et al., 2017b*), increased disorder was observed at the periphery of the complex in all structures possibly due to minor deviations from C4 symmetry. To see whether resolution of SUR1 could

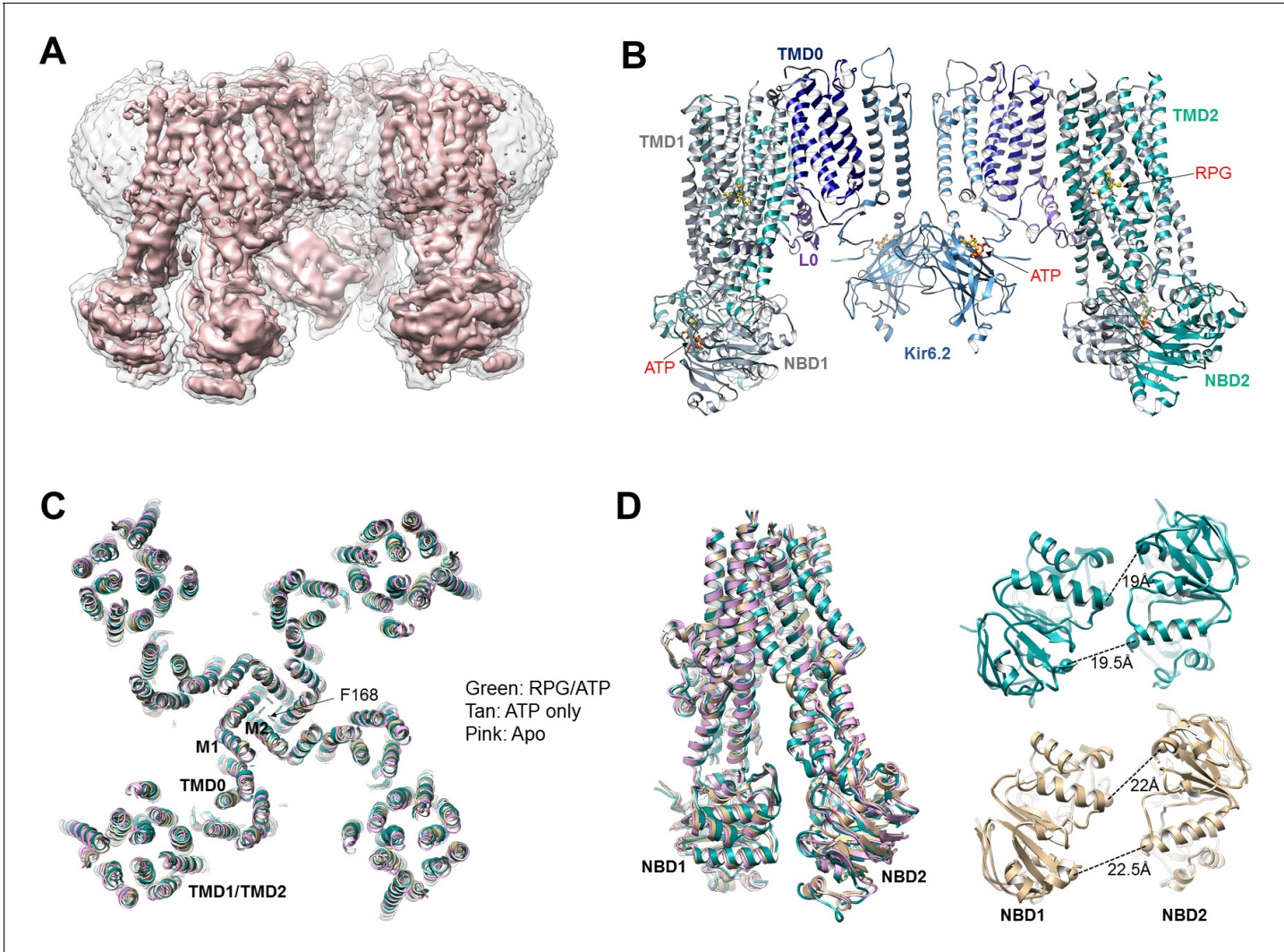

**Figure 1.** Structural determination and comparison. (**A**) Unsharpened 3.9 Å C4 cryoEM reconstruction of the $K_{ATP}$ channel bound to RPG and ATP. (**B**) Structural model of the channel in the RPG/ATP state. (**C**) Overlay of the RPG/ATP state structure, the ATP only state structure, and the apo state structure viewed from the top showing similarity of the dominant class of the ATP only and the apo state to the RPG/ATP state structure. (**D**) *Left*: Same model as (**C**) viewed from the side and focusing on the ABC transporter core module of SUR1 to illustrate the inward-facing conformation observed in all three structures. *Right*: Separation between Walker A and the signature motif in NBD1 and NBD2 (G716::S1483 and S831::G1382; Cα to Cα, indicated by the dashed line) in SUR1 bound to RPG and ATP (green) and ATP only (tan) viewed from the bottom.
DOI: https://doi.org/10.7554/eLife.46417.002

The following figure supplements are available for figure 1:

**Figure supplement 1.** Data collection and image processing workflow for the RPG/ATP state.
DOI: https://doi.org/10.7554/eLife.46417.003

**Figure supplement 2.** Data processing workflow for the CBZ/ATP state.
DOI: https://doi.org/10.7554/eLife.46417.004

**Figure supplement 3.** Data processing workflow for the GBC/ATP state.
DOI: https://doi.org/10.7554/eLife.46417.005

**Figure supplement 4.** Data processing workflow for the ATP only state and the apo state.
DOI: https://doi.org/10.7554/eLife.46417.006

**Figure supplement 5.** Comparison of the GBC/ATP state maps before and after focused refinement.
DOI: https://doi.org/10.7554/eLife.46417.007

**Table 1.** Statistics of cryo-EM data collection, 3D reconstruction and model building.

| Data collection | Rpg/ATP | Cbz/ATP | Gbc/ATP | ATP only | Apo |
|---|---|---|---|---|---|
| Microscope | Krios | Krios | Krios | Krios | Arctica |
| Voltage (kV) | 300 | 300 | 300 | 300 | 200 |
| Camera | Falcon III | Gatan K2 | Gatan K2 | Gatan K2 | Gatan K2 |
| Camera mode | Counting | Super-resolution | Super-resolution | Super-resolution | Super-resolution |
| Defocus range (μm) | −1.0 ~ −2.6 | −1.4 ~ −3.0 | −1.4 ~ −3.0 | −1.4 ~ −3.0 | −1.4 ~ −3.0 |
| Movies | 5765 | 4413 | 2180 | 2344 | 2047 |
| Frames/movie | 240 | 60 | 60 | 60 | 60 |
| Exposure time (s) | 60 | 15 | 15 | 15 | 15 |
| Dose rate (e⁻/pixel/s) | ~0.7 | 8 | 8 | 8 | 8 |
| Magnified pixel size (Å) | 1.045 | 1.72* | 1.72* | 1.71* | 1.826** |
| Total Dose (e-/Å2) | ~40 | ~40 | ~40 | ~40 | ~40 |
| **Reconstruction** | | | | | |
| _Whole channel_ | | | | | |
| Software | Relion 3.0 | Relion 2 | Relion 2 | Relion 2 | Relion 2 |
| Symmetry | C4 | C4 | C4 | C4 | C4 |
| Particles refined | 24,747 | 138,000 | 63,227 | 80,304 | 34,527 |
| Resolution (masked) | 3.9 Å | 4.4 Å | 4.07 Å | 4.88 Å | 5.31 Å |
| _SUR1 focused refinement_ | | | | | |
| Software | Relion 3.0 | Relion 3.0 | Relion 3.0 | Relion 3.0 | Relion 3.0 |
| Symmetry | C1 | C1 | C1 | C1 | C1 |
| Particles refined | 312,000 | 499,095 | 171,420 | 123,757 | 90,058 |
| Resolution (masked) | 3.65 Å | 4.34 Å | 3.74 Å | 4.5 Å | 4.55 Å |
| **Model Statistics** | (Includes KNt) | | (Includes KNt) | | |
| Map CC (masked) | 0.6559 | 0.7771 | 0.7065 | 0.8155 | 0.7677 |
| Clash score | 3.37 | 7.52 | 3.22 | 2.60 | 3.11 |
| Molprobity score | 1.58 | 1.95 | 1.60 | 1.46 | 1.5 |
| Cβ deviations | 0 | 0 | 0 | 0 | 0 |
| **Ramachandran** | | | | | |
| Outliers | 0% | 0% | 0% | 0% | 0% |
| Allowed | 7.03% | 9.66% | 7.98% | 6.29% | 5.93% |
| Favored | 92.97% | 90.34% | 92.02% | 93.71% | 94.07% |
| **RMS deviations** | | | | | |
| Bond length | 0.009 | 0.008 | 0.007 | 0.005 | 0.009 |
| Bond angles | 1.166 | 1.121 | 0.992 | 1.085 | 1.233 |

*Super-resolution pixel size 0.86; **Super-resolution pixel size 0.913.

DOI: https://doi.org/10.7554/eLife.46417.008

be improved, focused refinement using symmetry expansion and signal subtraction with different masks was performed on all structures, including the previously collected GBC/ATP dataset (_Martin et al., 2017a_), as described in Materials and methods. This yielded maps with improved reported resolutions for SUR1 as follows: 3.74 Å for the GBC/ATP state, 3.65 Å for the RPG/ATP state, 4.34 Å for the CBZ/ATP state, 4.50 Å for the ATP-only state, and 4.55 Å for the apo state using the FSC cutoff of 0.143 (_Table 1_).

For model building, we used our previously published GBC/ATP structure (PDB: 6BAA) (_Martin et al., 2017a_) as a starting point and refined the models against the experimental data. For the GBC/ATP structure, the new map derived from focused refinement of the Kir6.2 tetramer plus

an SUR1 subunit showed significantly improved cryoEM density in a number of regions of SUR1 previously not modeled in the GBC/ATP map (see deposited PDB and EMD files). This allowed us to build additional residues into the GBC/ATP structure. In particular, the ATP density in NBD1 became clearly resolved (*Figure 1—figure supplement 5A*; note ATP binds NBD1 in the absence of $Mg^{2+}$[*Ueda et al., 1997*]). The linker between TMD2 and NBD2 (L1319-Q1342) also showed much improved density, allowing us to build a continuous polyalanine model (*Figure 1—figure supplement 5B*). Models for the other structural states were similarly built according to the highest resolution maps available for the focus-refined regions (for details see Materials and methods).

In all structures determined in the presence of pharmacochaperones and ATP, the Kir6.2 tetramer is in a closed conformation (*Figure 1C*), and the SUR1's ABC core in an 'inward-facing' conformation wherein the NBD1 and NBD2 are separated (*Figure 1D*). This overall structure is very similar to that previously reported for the GBC/ATP state (*Martin et al., 2017a*; *Martin et al., 2017b*). Interestingly, the dominant class emerging from 3D classification for the ATP-only state as well as the apo state presented a similar conformation, with Kir6.2 tetramer closed and SUR1 inward-facing (*Figure 1C and D*). Of note, a dominant inward-facing conformation in the absence of ligand was also observed in another ABC transporter, the multidrug resistance protein MRP1 (*Johnson and Chen, 2017*). These observations suggest the closed channel conformation is the most stable for the apo state under our experimental conditions (i.e. no ATP and no exogenous $PIP_2$, a lipid required for $K_{ATP}$ channel opening [*Nichols, 2006*]). More data and extensive analyses will be needed to resolve whether other minor conformations are present in our samples and to understand the dynamics of these structures, especially in the ATP-only and the apo states. Here, we focus on analyzing the pharmacochaperone binding pocket and the mechanisms by which pharmacochaperones affect channel assembly and activity.

## RPG and CBZ are located in the GBC binding pocket

We previously solved the $K_{ATP}$ structure in the presence of GBC and ATP, revealing that GBC is lodged in the TM bundle above NBD1 (*Martin et al., 2017a*), a finding which has been independently confirmed (*Wu et al., 2018*). Here, in both the RPG/ATP and CBZ/ATP structures, we observe strong and distinctly shaped cryo-EM densities within the same GBC binding pocket (*Figure 2B–D*). Such density is absent in the ATP-only and the apo structures (*Figure 2E and F*), supporting assignment as the pharmacochaperone ligand.

Density for RPG appears compact and palm-shaped, and suggests that the molecule adopts a considerably folded shape upon binding to SUR1 (*Figure 2B*; *Figure 2—figure supplement 1B*). Interestingly, RPG possesses a carboxyl group adjacent a benzyl group, analogous to the sulfonyl group in GBC that is also adjacent a benzyl group (*Figure 2G*). Refinement of an RPG molecule into the observed binding pocket density orients this carboxylate towards N1245, R1246, and R1300 (*Figure 3A*), which coordinate the sulfonyl group in the GBC-bound structure (*Figure 3B*). However, unlike GBC, the RPG density is distant from S1238. This explains previous functional data that an SUR1 S1238Y mutation does not affect RPG's ability to modulate channel function (*Hansen et al., 2002*; *Yan et al., 2006*). The helix on the opposite side of the binding pocket (i.e. TM8; *Figure 3A*) is lined with hydrophobic residues (W430, F433, and L434), which may support binding through a combination of van der Waals interactions and shape complementarity. Interestingly, although similar in overall structures, we noted subtle rearrangements within the binding pocket between GBC- and RPG-bound states (*Figure 3A and B*). The most obvious is in W1297, which in RPG is flipped down towards the ligand. Thus, there is sufficient flexibility of the binding pocket to accommodate diverse compounds with high affinity.

CBZ is a smaller molecule with molecular weight about half of that of GBC and RPG (*Figure 2G*). However, in the CBZ-bound SUR1 structure the cryoEM density corresponding to the ligand has a size and shape that markedly resembles GBC (*Figure 2C and D*), with one end pointing towards S1238. While the close proximity to S1238 is in agreement with our previous finding that mutation of S1238 to Y diminishes the ability of CBZ to both inhibit and chaperone the channel (*Devaraneni et al., 2015*), the density is too large to be fitted by a single CBZ molecule (*Figure 2C*; *Figure 2—figure supplement 1C*). The structure of CBZ has been extensively studied and multiple polymorphic crystalline forms have been reported, including dimers (*Florence et al., 2006*; *Grzesiak et al., 2003*). Thus, an intriguing possibility is that CBZ may bind as a dimer to occupy the entire binding pocket (*Figure 2—figure supplement 1C*). Alternatively, the result could be

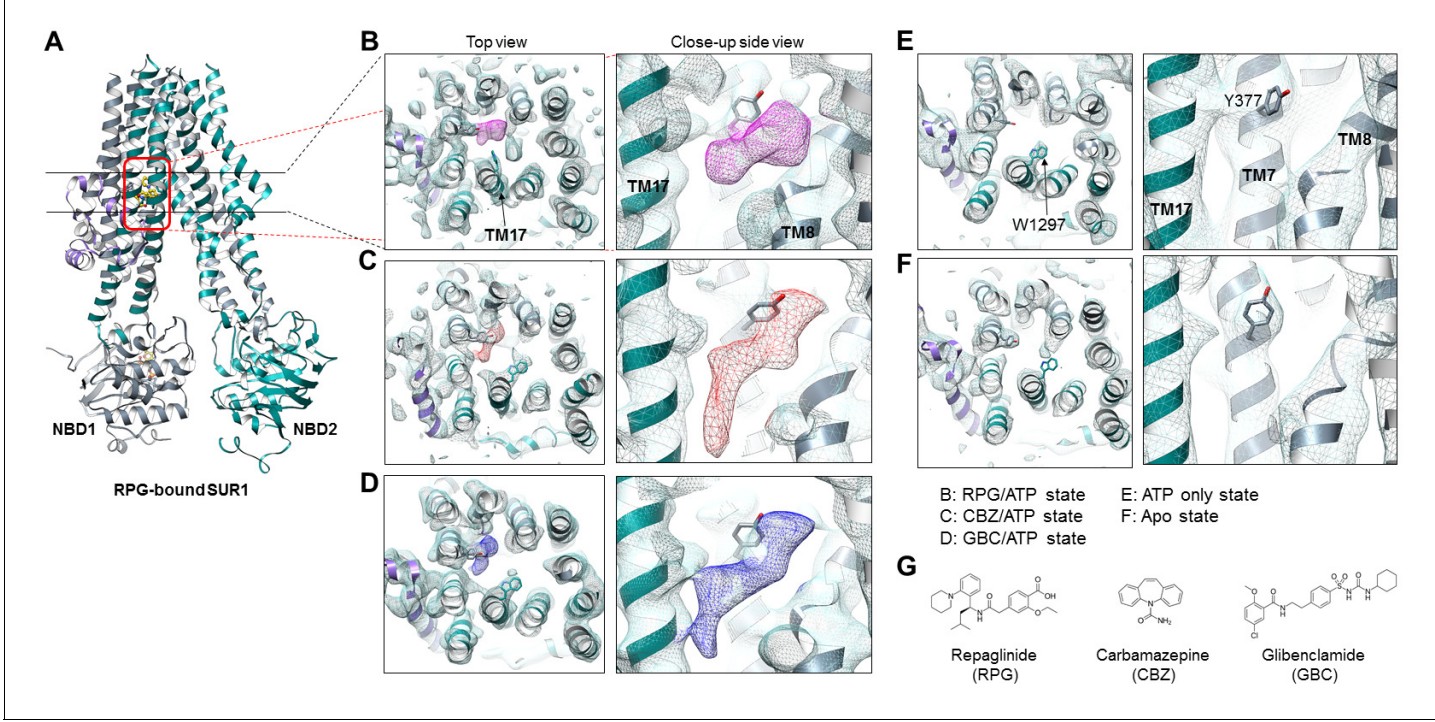

**Figure 2.** Structural comparison of the pharmacochaperone binding pocket. (**A**) Structural model of the RPG-bound SUR1 ABC transporter core module viewed from the side showing the slice viewed from the top (indicated by the two black lines) and the pocket viewed from the side at higher magnification (indicated by the red box) in B-F. (**B–F**) The pharmacochaperone pocket viewed from the top and the side of the channel in the states indicated. To enable comparison, each map was sharpened and filtered to 4.6 Å (the resolution of the apo state reconstruction) with the Postprocessing procedure in RELION. Ligand density corresponding to RPG in (**B**) is shown in magenta, CBZ in (**C**) in red, and GBC in (**D**) in blue. The binding pocket is empty in both the ATP only state (**E**) and the apo state (**F**). Note the side chain of W1297 in TM17 and Y377 in TM7 are shown and labeled in panel (**E**) to serve as reference points. (**G**) Chemical structures of the three pharmacochaperones shown in **B–D**.

DOI: https://doi.org/10.7554/eLife.46417.009

The following figure supplement is available for figure 2:

**Figure supplement 1.** Density fitting for GBC, RPG, and CBZ.
DOI: https://doi.org/10.7554/eLife.46417.010

explained by a CBZ molecule having multiple occupancies, and what is observed in the reconstruction is an average of the ensemble. Due to this uncertainy we did not model the CBZ molecule in the structure. To our knowledge this is the first protein structure determined in complex with CBZ. Thus, there are no structures for comparison. More studies are needed to distinguish these possibilities.

Structural evidence above indicates that the three pharmacochaperones occupy a common binding pocket to exert their effects. To seek functional evidence, we mutated five select SUR1 residues lining the binding pocket to Ala (*Figure 3A*) and monitored the effect of mutation on the ability of GBC, CBZ and RPG to correct the trafficking defects in SUR1-TMD0 mutants. The trafficking mutation F27S in SUR1 has been well-characterized in previous work, showing nearly undetectable mature complex-glycosylated form (upper band in *Figure 3C*) in the absence of pharmacochaperones but strong recoveries with both CBZ and GBC (*Chen et al., 2013*). In an F27S background, we introduced the five binding pocket mutations which have been shown previously to not significantly alter SUR1 maturation by themselves (*Martin et al., 2017a*), and examined the impact on the chaperone function of CBZ, GBC and RPG. In Western blots, there was a complete absence of the mature upper band for F27S-SUR1 in vehicle (DMSO) treated control. In contrast, there were strong upper bands when the mutant was expressed in the presence of CBZ, RPG or GBC (*Figure 3C*). This chaperone ability of the drugs was reduced to a variable extent for each of the binding site mutations tested in the F27S background (*Figure 3C*). Interestingly, we noted that while the sensitivity profile is very similar for GBC and CBZ, that for RPG is slightly different, in particular to the R1246A

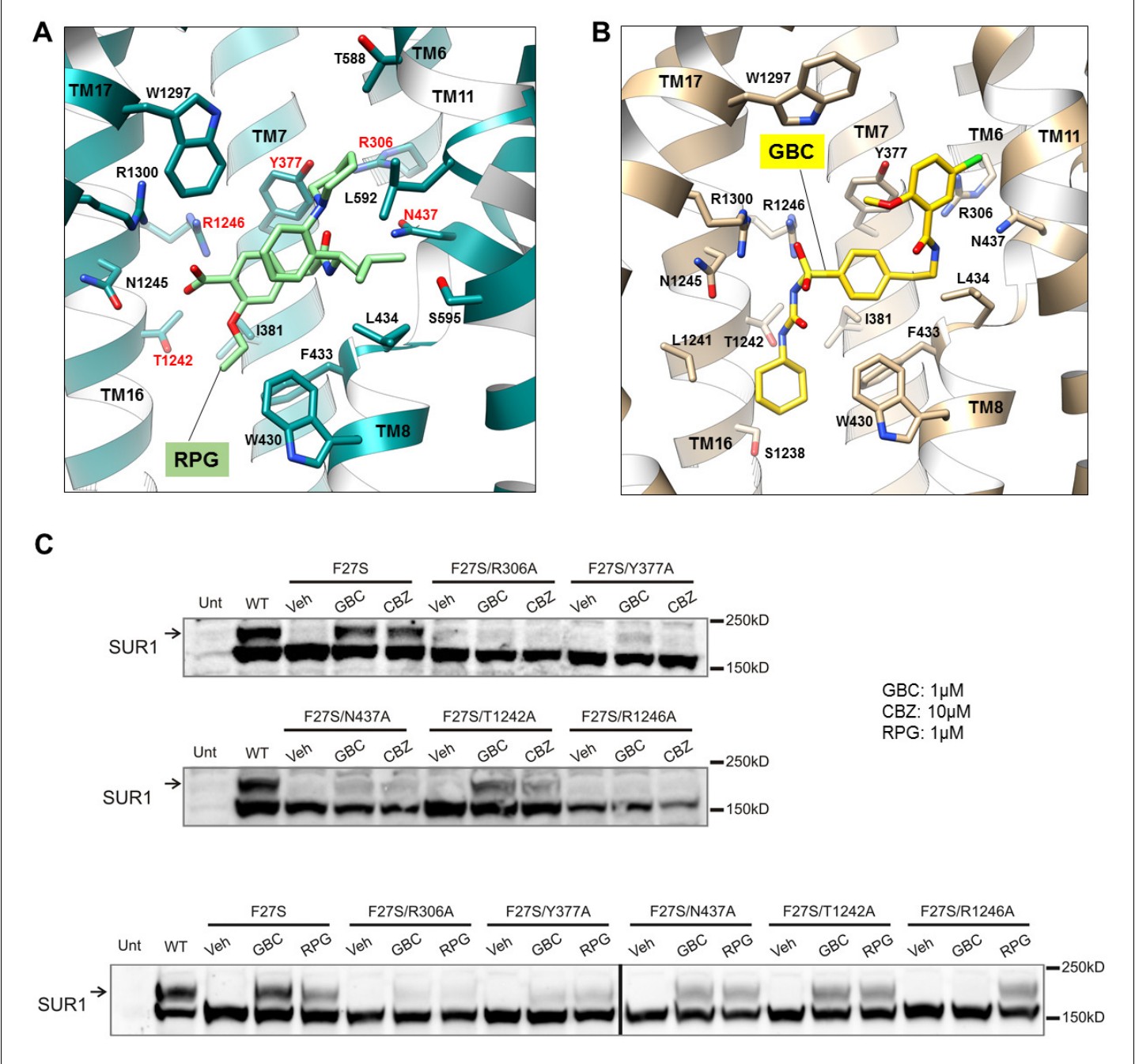

**Figure 3.** Models of the PC binding pocket. (**A**) RPG binding site model, with residues mutated to alanine in C labeled in red. (**B**) GBC binding site model. (**C**) Western blots showing effects of alanine mutation of the selected residues on the ability of GBC, CBZ, and RPG to correct the processing defect caused by the F27S mutation in the TMD0 of SUR1. The arrow indicates the mature, complex-glycosylated SUR1. The lower band is the core-glycosylated immature SUR1. The thick vertical line in the bottom blot indicates samples from the same experiments run on two separate gels.
DOI: https://doi.org/10.7554/eLife.46417.011

mutation. This suggests despite sharing a common binding pocket, RPG forms distinct chemical interactions with surrounding residues compared to GBC and CBZ. The combined structural and functional data presented above provide strong evidence that CBZ, RPG, and GBC exert their pharmacochaperoning effects by binding to a common binding pocket we have here identified.

## Structure of the distal N-terminus of Kir6.2

The N-terminal ~30 amino acids of Kir6.2 (referred to as KNt hereinafter) has been known to be critical for channel assembly, gating, and interaction with sulfonylureas and glinides. However, underlying mechanisms remain poorly understood. Early studies showed that deletion of KNt significantly increases channel open probability (*Babenko et al., 1999*; *Koster et al., 1999b*; *Reimann et al., 1999*; *Shyng et al., 1997*), reduces ATP inhibition, and renders the channel less sensitive to sulfonylureas (*Koster et al., 1999a*; *Reimann et al., 1999*). KNt also appears to contribute to GBC binding (*Kühner et al., 2012*; *Vila-Carriles et al., 2007*), and is necessary for high affinity interaction with RPG (*Hansen et al., 2005*; *Kühner et al., 2012*). Deletion of amino acids 2–5 from the KNt shifts the binding affinity of RPG by more than 30-fold (*Kühner et al., 2012*). Moreover, KNt is critical for channel assembly and pharmacochaperoning (*Devaraneni et al., 2015*; *Schwappach et al., 2000*). Removal of KNt markedly reduces the ability of GBC and CBZ to rescue SUR1-TMD0 trafficking mutations (*Devaraneni et al., 2015*). Our recent studies showing that *p*-azidophenylalanine genetically incorporated at Kir6.2 amino acid position 12 or 18 was photocrosslinked to SUR1 and that the extent of crosslinking increased in the presence of GBC or CBZ further suggest physical interactions between KNt and SUR1 in a drug-sensitive manner (*Devaraneni et al., 2015*). Taken together, these studies led us to hypothesize that KNt is located near the pharmacochaperone binding pocket we have identified.

Close examination of the three pharmacochaperone-bound SUR1 structures reconstructed using focused refinement indeed revealed significant and continuous cryo-EM density immediately adjacent to the drug binding site, especially in the RPG-bound map (*Figure 4A and B*). This density appears as a roughly linear peptide inserting between the two TMDs of SUR1 from the intracellular side. By filtering the GBC/ATP and RPG/ATP maps to a lower resolution (6 Å), we were able to see contiguous density that extends out of the SUR1's central cavity and connects to the density of the first structured residue R32 in Kir6.2 (*Figure 4—figure supplement 1*). Of note, a recent report by Wu et al. also pointed out a density in the SUR1's central cavity that is nearly contiguous with the Kir6.2 cytoplasmic domain in our published GBC-bound structure and proposed it to be the N-terminus of Kir6.2 (*Wu et al., 2018*) (see Discussion).

Interestingly, in the RPG-bound structure the piperidino moiety of RPG is in close proximity to the density that corresponds to the most N-terminus of the KNt (*Figure 4—figure supplement 2A*). This observation agrees with previous findings that the increased affinity to RPG caused by Kir6.2 is due to the peperidino group (*Stephan et al., 2006*), and that deletion of only a few amino acids from the Kir6.2 N-terminus decreases RPG binding affinity by more than 30-fold (*Kühner et al., 2012*). Of note, the KNt density is also seen in the ATP-only reconstruction though not as strong or well-defined as in the drug-bound reconstructions (i.e., the density disappears at lower σ values for the ATP-only map), but is largely absent in the apo state structure (*Figure 4C*). This indicates KNt enters into the central cavity of the inward-facing SUR1 even in the absence of drugs, and that drug binding stabilizes the location of KNt in the central cavity. We found that deletion of amino acids 2–5 or 2–10 increasingly reduced the ability of the pharmacochaperones to correct the processing defect of the F27S TMD0 mutant SUR1 (*Figure 4—figure supplement 2B*). Although the reduction in the F27S-SUR1 mature band intensity in these experiments could be due to a destabilization effect of KNt deletions on F27S mutant SUR1, it could also suggest that the KNt is needed for the pharmacochaperoning effect. Interestingly, the same Kir6.2 N-terminal deletion mutations also reduced the maturation of WT SUR1 and the ability of pharmacochaperones to enhance WT SUR1 maturation (*Figure 4—figure supplement 2C*). These structural and functional observations lead us to propose that KNt binds to the central cavity of the SUR1 ABC core and stabilizes SUR1-Kir6.2 association by simultaneous interaction with multiple SUR1 TMD helices to facilitate complex assembly. This mechanism not only is important for efficient biogenesis of WT channels but also likely underlies the ability of pharmacochaperones to enhance WT channel biogenesis efficiency and rescue mutant channel assembly (see Discussion).

## Physical and functional interactions of the distal N-terminus of Kir6.2 with SUR1

While our cryoEM structures support the assignment of the KNt density, which is consistent with the many functional experiments described above (*Babenko and Bryan, 2002*; *Babenko et al., 1999*;

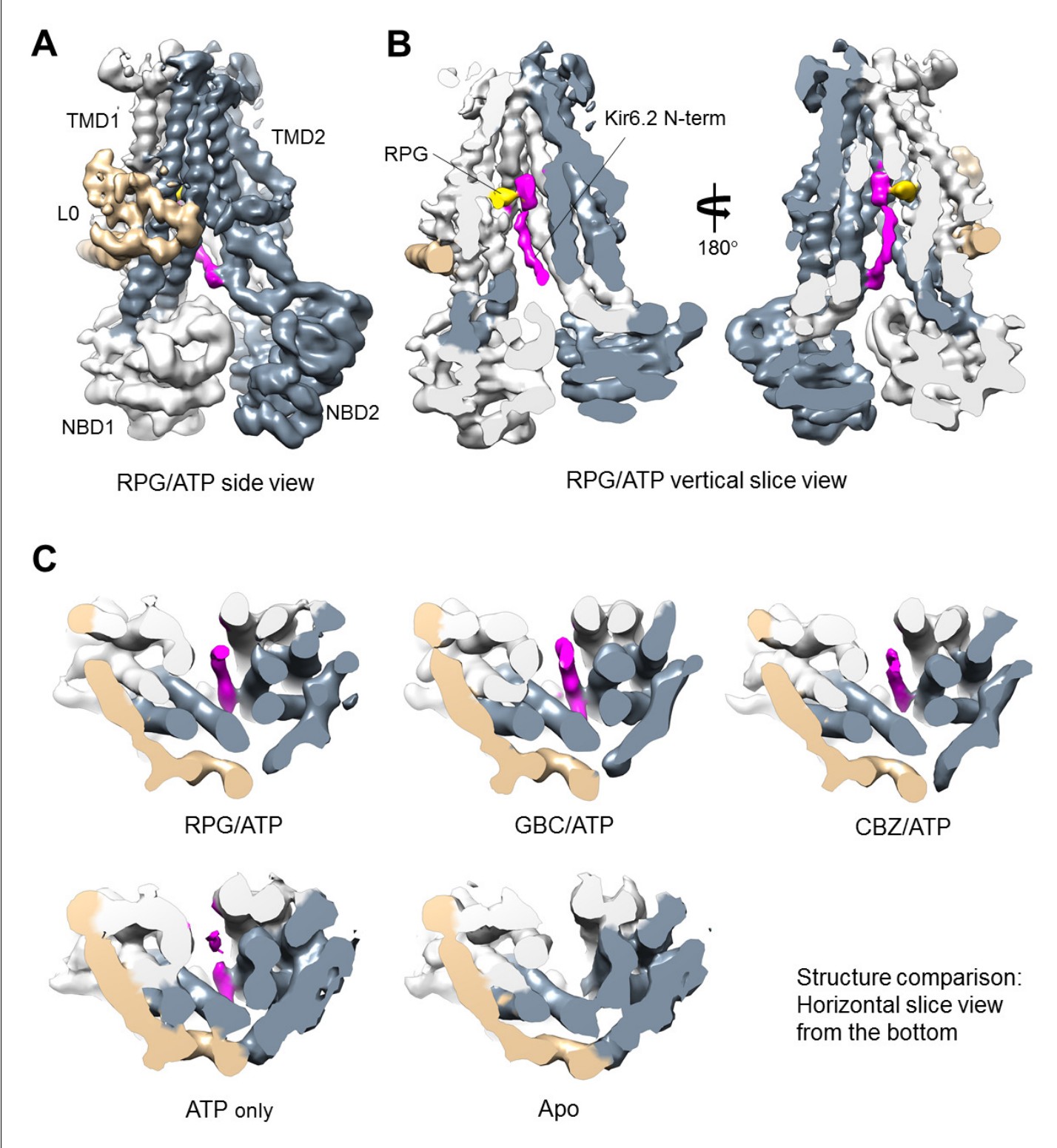

**Figure 4.** Kir6.2 N-terminus cryoEM density in SUR1. (**A**) RPG-bound SUR1 from focus-refined, unsharpened map viewed from the side. The major domains are labeled in different colors. (**B**) Vertical slice view of the map shown in (**A**) that reveals the bound RPG (in gold) and the cryoEM density of Kir6.2 N-term (in magenta). (**C**) Comparison of the Kir6.2 N-term cryoEM density in the different structures in horizontal slices viewed from the bottom. All maps are sharpened and filtered to 6 Å. Apo and ATP-only structures are displayed at 1.8σ; RPG/ATP, GBC/ATP, and CBZ/ATP structures are displayed at 2.2σ. Note lower threshold is needed for the Kir6.2 N-term density in the ATP-only structure to become visible.

DOI: https://doi.org/10.7554/eLife.46417.012

*Figure 4 continued on next page*

*Figure 4 continued*

The following figure supplements are available for figure 4:

**Figure supplement 1.** CryoEM density of Kir6.2 N-terminus in the GBC/ATP and RPG/ATP structures.

DOI: https://doi.org/10.7554/eLife.46417.013

**Figure supplement 2.** The distal N-terminus of Kir6.2 interacts with SUR1 and is required for channel biogenesis and pharmacochaperone rescue.

DOI: https://doi.org/10.7554/eLife.46417.014

**Figure supplement 3.** Mass spectrometric identification of a chemical crosslink between Kir6.2 peptide KGIIPEEYVLTR (5-16) and SUR1 peptide STVKALVSVQK (599-609) using CBDPS.

DOI: https://doi.org/10.7554/eLife.46417.015

*Devaraneni et al., 2015*; *Koster et al., 1999b*; *Martin et al., 2016*), direct physical evidence linking KNt to SUR1 residues lining the central cavity is still lacking. Identification of crosslinked SUR1 residue via *p*-azidophenylalanine engineered in KNt is technically challenging due to the complex chemistry of the azido-mediated crosslinks (*Devaraneni et al., 2015*). As an alternative approach, we performed crosslinking experiments using an amine-reactive homobifunctional crosslinker CBDPS (Cyanurbiotindimercaptopropionylsuccinimide) with purified $K_{ATP}$ channels bound to GBC, followed by mass spectrometry to identify crosslinked peptides, as described in Materials and methods. One of the inter-SUR1-Kir6.2 crosslinks we identified connected Kir6.2 lysine five and SUR1 lysine 602 (*Figure 4—figure supplement 3*), which is near the drug binding site in our structural model. The distance between α carbons (Cα) of the crosslinked lysines in our model is ~18 Å, which is within the reported range for the linker (*Brodie et al., 2019*). This result provides the first evidence for the physical proximity of KNt to the SUR1 ABC core central cavity.

Because the crosslinking/mass spectrometry experiments were performed using purified channels in detergents, there is a possibility that the interaction we have identified does not occur in cell membranes under physiological condition. To further corroborate the crosslinking/mass spectrometry results and focus on the interaction, we engineered cysteine pairs guided by the structure to test whether KNt can be crosslinked to SUR1. Inspection of our structural model pointed to an endogenous cysteine C1142 in SUR1 with proximity to the distal end of KNt (*Figure 4—figure supplement 2A*). Accordingly, we mutated Kir6.2 amino acids 2–5 to cysteine and asked whether the engineered Kir6.2 cysteines formed disulfide bonds with SUR1-C1142. We hypothesized that crosslinking of KNt with SUR1-C1142 would trap KNt within the SUR1 ABC core and reduce channel open probability, based on previous studies that deletion of KNt increases channel open probability (*Babenko and Bryan, 2002*; *Babenko et al., 1999*; *Koster et al., 1999a*; *Reimann et al., 1999*; *Shyng et al., 1997*).

Using inside-out patch-clamp recording, channel activity was monitored in response to the oxidizing reagent $I_2$ followed by exposure to the reducing reagent dithiothreitol (DTT) (*Anishkin et al., 2008*). We tested WT channels, channels formed by WT SUR1 (C1142) and Kir6.2-L2C, S3C, R4C or K5C, as well as channels formed by SUR1-C1142A and Kir6.2-L2C, S3C, R4C, or K5C. Current decay was observed to a variable extent in most patches even in bath solution without $I_2$. This behavior, referred to as rundown is typical of $K_{ATP}$ channels and is attributed to loss of $PIP_2$ (*Lin et al., 2003*; *Ribalet et al., 2000*). Channel rundown tends to plateau and is not spontaneously reversible. Because of rundown, it is difficult to discern current decay caused by $I_2$-induced crosslinking; however, the current decay component caused by crosslinking should be reversible following exposure to DTT. Strikingly, co-expression of Kir6.2-L2C with WT SUR1 resulted in channels that displayed current recovery upon exposure to DTT (10 mM) following the initial exposure to $I_2$ (250 μM). Importantly, none of the other Kir6.2 cysteine mutations showed DTT-induced current recovery when co-expressed with SUR1-C1142 (R4C shown as an example in *Figure 5B*). WT channels or channels formed by co-expressing Kir6.2-L2C and a SUR1 in which C1142 was mutated to an alanine (C1142A) also failed to show current reversal in DTT. Note not all patches of channels formed by WT SUR1 and Kir6.2-L2C showed significant current recovery in DTT; these patches also had less current decay in $I_2$ (*Figure 5E*). It is possible that these are deep patches where solution exchange was slower; alternatively we speculate that $PIP_2$ loss might be slower in these patches such that KNt was for the most part not inserted in SUR1's central cavity to be crosslinked. Regardless, our observations that more than half of the WT-SUR1/Kir6.2-L2C patches (8 out of 14) showed significant DTT-induced current recovery while none of the other channels tested did (>40 patches) support

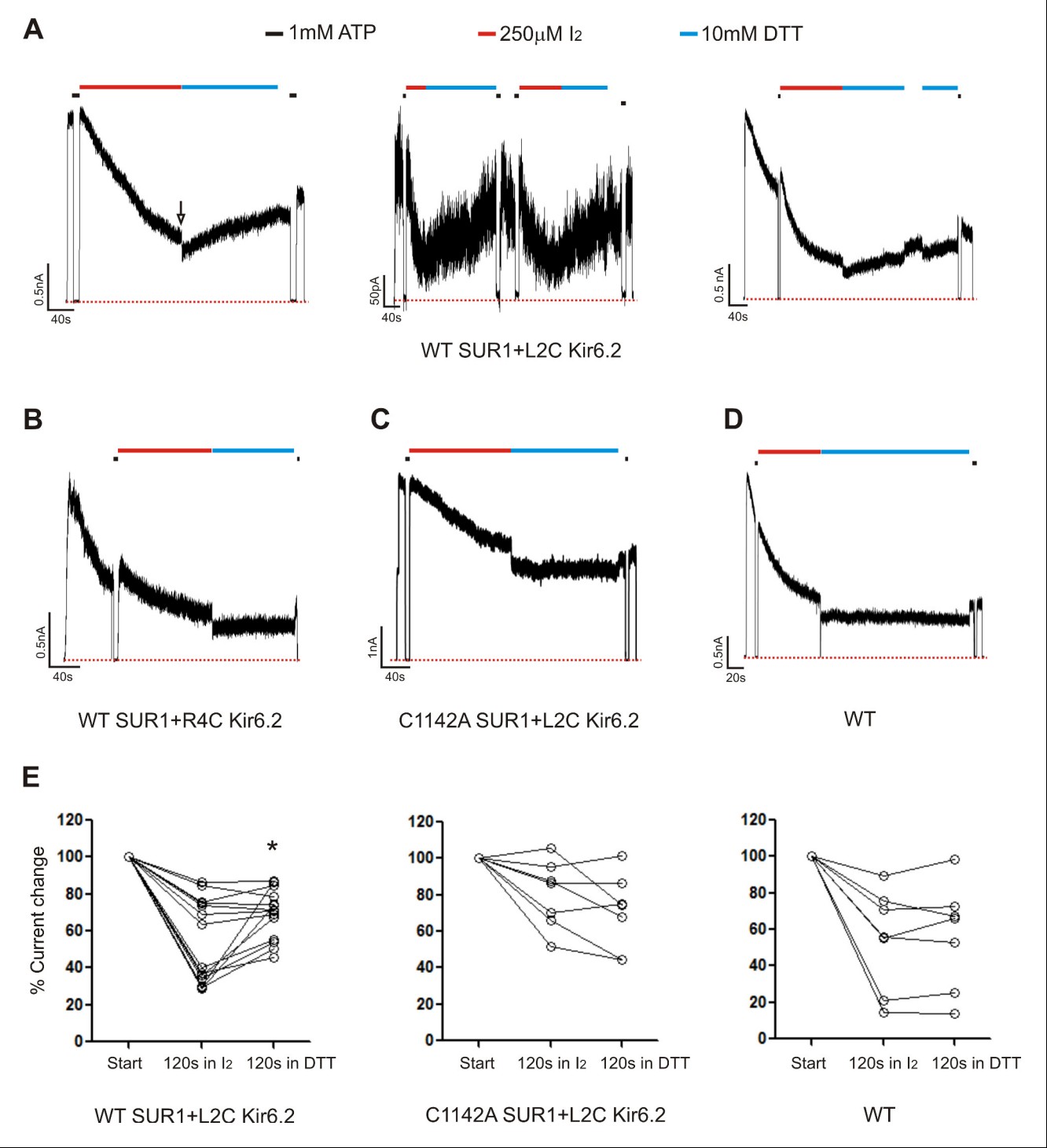

**Figure 5.** Patch-clamp recording of cysteine mutants to probe location of the Kir6.2 N-terminus. (**A**) Three examples of traces of channels formed by WT(C1142) SUR1 and L2C Kir6.2 showing DTT-induced current recovery following $I_2$ exposure. Channels were exposed to 250 μM $I_2$ or 10 mM DTT as indicated by the bars above the traces. Baseline was obtained by exposing channels to 1 mM ATP. The arrow in the first trace indicates the blocking effect of the DTT that was readily reversed upon return to K-INT. Note as we have documented before (*Lin et al., 2003*; *Pratt et al., 2012*), DTT alone did not cause significant current run-up or run-down aside from the reversible blocking effect (not shown). (**B, C, D**) Representative traces of control channels formed by WT SUR1 and R4C Kir6.2, C1142A SUR1 and L2C Kir6.2, or WT SUR1 and Kir6.2. (**E**) Quantification of current changes at the end of 120 s exposure to $I_2$, and at the end of subsequent 120 s exposure to DTT (expressed as % of currents at the start of $I_2$ exposure). The asterisk indicates statistical significance comparing currents at the end of $I_2$ exposure and the end of DTT exposure using paired student's *t*-test ($p < 0.05$).
DOI: https://doi.org/10.7554/eLife.46417.016

crosslinking between SUR1-C1142 and Kir6.2-L2C. Together our structural and functional data is consistent with the idea that insertion of KNt into the central cavity of SUR1 ABC core serves as a mechanism to reduce channel open probability, in addition to promoting channel assembly.

## Discussion

Loss of membrane protein expression due to impaired folding and trafficking underlies numerous human diseases. In some cases, such defects can be overcome by small molecule ligands that bind to mutant proteins, referred to as pharmacochaperones (*Bernier et al., 2004*; *Hanrahan et al., 2013*; *Powers et al., 2009*; *Ringe and Petsko, 2009*). Although significant progress has been made in discovering pharmacochaperones, as exemplified for the CFTR mutation ΔF508 (*Mijnders et al., 2017*; *Pedemonte et al., 2005*; *Veit et al., 2018*), detailed structural mechanisms of how these compounds work are in most cases not well understood. This problem is even more complex in hetero-multimeric proteins such as the $K_{ATP}$ channel where not only folding of the mutant protein itself but interactions with assembly partners may be involved. In this study, we set out to understand how structurally diverse compounds that inhibit $K_{ATP}$ channels act as pharmacochaperones to correct channel trafficking defects. We present structural, biochemical, and functional evidence that GBC, RPG, and CBZ, despite diverse chemical structures, occupy a common pocket in SUR1, that this occupancy stabilizes a labile binding interaction between the N-terminus of Kir6.2 and the central cavity of the SUR1 ABC core, and that this interaction controls channel activity as well as the assembly and trafficking of nascent channels to the β-cell surface. Our findings shed light on the principles that govern the assembly of the $K_{ATP}$ complex and offer insight into the mechanism by which channel inhibitors act allosterically to rescue channel trafficking defects caused by SUR1 mutations in TMD0 (see *Figure 6*).

Among ABC transporters and inward rectifier potassium channels, SUR1 and Kir6.2 are unique in having evolved to become functionally inter-dependent. The mechanisms by which these two structurally unrelated proteins form a heteromeric complex to regulate $K^+$ transport and insulin secretion have been intensively investigated. Recently, several cryo-EM structures have been reported of $K_{ATP}$ channels naturally assembled from separate SUR1 and Kir6.2 proteins (*Li et al., 2017*; *Martin et al., 2017a*; *Martin et al., 2017b*), or comprising engineered SUR1(C)-(N)Kir6.2 fusion proteins (*Lee et al., 2017*; *Wu et al., 2018*). These structures reveal the channel's overall architecture as a Kir6.2 tetramer surrounded by four SUR1, wherein Kir6.2 makes direct contact with the TMD0 of SUR1. This structure explains why numerous trafficking mutations are found in TMD0 (*Snider et al., 2013*). However, we have shown in the structure of GBC-bound $K_{ATP}$ complexes that, GBC resides well within the TM bundle of the SUR1 ABC core, above NBD1 and distant from TMD0 (*Martin et al., 2017a*). An allosteric mechanism by which binding of GBC rescues TMD0 trafficking mutations has remained obscure.

A critical missing piece of the puzzle was the KNt, which we have shown to be involved in pharmacochaperone rescue of TMD0 mutations (*Devaraneni et al., 2015*). KNt (Kir6.2 aa 1–31) was not well resolved in the initial published GBC-bound structures (*Li et al., 2017*; *Martin et al., 2017a*; *Martin et al., 2017b*). Recently, Wu et al. compared a GBC-bound cryoEM channel structure using a SUR1-Kir6.2 fusion construct with a 39aa linker to previously published GBC-bound structures of channels formed by co-expression of SUR1 and Kir6.2 as separate proteins (*Wu et al., 2018*). They noted a cryoEM density in SUR1's central cavity that projects towards the Kir6.2 cytoplasmic domain in the SUR1 and Kir6.2 co-assembled structures but not the SUR1-Kir6.2 fusion structure. Despite this density not being contiguous with the structured Kir6.2 N-terminus, it was reasoned the density likely corresponds to KNt based on previous structure-function data. The authors argue that the lack of density in the fusion cryoEM map is likely due to the large size of the linker that cannot be accommodated in the central cavity. Here, through improved focused refinement algorithms and cross-correlation of multiple structures we were able to discern a low-resolution, contiguous cryoEM density emanating from the structured Kir6.2 R32 and ending near the top of the SUR1's central cavity, allowing us to assign the KNt density with confidence. This assignment is further supported by the crosslinking/mass spectrometry data connecting Kir6.2-K5 to SUR1-K602. Interestingly, the crosslinking/mass spectrometry experiment also identified another crosslink between Kir6.2-K5 and SUR1-K205 (*Figure 4—figure supplement 3*). That Kir6.2-K5 is able to interact with both SUR1-K602 lining

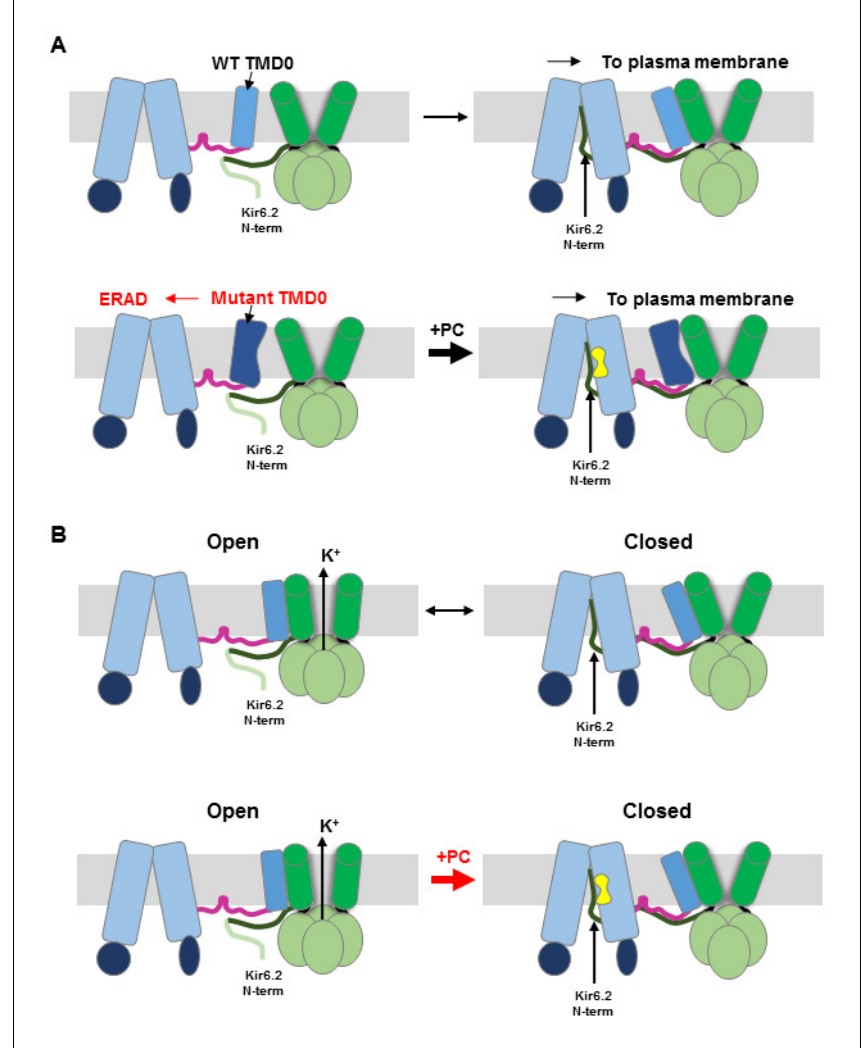

**Figure 6.** Cartoon of pharmacochaperoning and channel inhibition mechanism. (**A**) *Top:* During WT channel biogenesis, KNt insertion into the SUR1 central cavity facilitates SUR1-TMD0 interaction with Kir6.2 for successful channel assembly. For SUR1 containing TMD0 trafficking mutations, it is unable to assemble with Kir6.2 in the absence of a pharmacochaperone (PC), and is targeted for ER-associated degradation (ERAD). Upon binding of PC to the binding pocket in SUR1, the Kir6.2 N-term becomes stabilized in the central cavity of the SUR1 ABC core, allowing assembly of the mutant TMD0 with Kir6.2 and trafficking of the complex to the plasma membrane. (**B**) Insertion of KNt into SUR1's central cavity prevents Kir6.2 from adopting an open conformation (top) to close the channel. However, this interaction is labile and reversible (indicated by the black arrow going both directions). Binding of PC in the SUR1 pocket stabilizes Kir6.2 N-term in the central cavity and pushes channel equilibrium towards a closed state (indicated by the thick red arrow pointing to the closed state). In addition, PC binding stabilizes SUR1 in an inward-facing conformation, unable to be stimulated by Mg-nucleotides. Crosslinking of SUR1's endogenous C1142 with engineered Kir6.2-L2C also traps the Kir6.2 N-term in the central cavity, closing the channel (see *Figure 5*).

DOI: https://doi.org/10.7554/eLife.46417.017

the central cavity and K205 in L0 is consistent with flexibility of KNt as reflected by the low cryoEM resolution of this stretch of amino acids.

The combined structural and functional data presented in this study provide a mechanism of pharmacochaperoning wherein pharmacochaperone binding acts in trans to overcome assembly defects caused by mutations within TMD0. Because the pharmacochaperone binding site is formed by the TM helices of the SUR1 ABC core, and TMD0 is a separate domain, it is unlikely that its occupancy by chemically diverse compounds overcome or prevent misfolding of TMD0 trafficking

mutations at the site of the mutation. Supporting this notion, GBC, RPG, and CBZ fail to chaperone TMD0 trafficking mutants in the SUR1$_{RKR-AAA}$ background, a variant in which the RKR ER retention signal has been mutated to AAA to allow Kir6.2-independent trafficking to the cell surface (*Devaraneni et al., 2015*; *Yan et al., 2006*). Rather, occupancy of the pharmacochaperone binding site stabilizes the insertion of KNt into the central cavity of the SUR1 ABC core, an otherwise labile interaction that is nonetheless crucial for assembly and trafficking of the channel complex out of the ER (*Figure 6A*). Progressive deletion of KNt (2-5, 2-10) renders pharmacochaperones less and less effective in rescuing the trafficking mutant (*Figure 4—figure supplement 2B*), likely due to progressive weakening of pharmacochaperone-dependent KNt-SUR1 interactions. However, we cannot rule out the possibility that deletion of Kir6.2 KNt further destabilizes the SUR1 trafficking mutant such that the mutant SUR1 can no longer be rescued by pharmacochaperones. Importantly, progressive deletion of KNt (2-5, 2-10) also markedly reduces the biogenesis efficiency of WT channels and makes pharmacochaperones less and less effective in enhancing the biogenesis efficiency of the channel (*Figure 4—figure supplement 2C*). These results lead us to propose insertion of KNt into the central cavity of the SUR1-ABC core as a principal mechanism to stabilize SUR1-Kir6.2 interaction during channel assembly. We suggest that during normal channel assembly, insertion of KNt into the SUR1 central cavity facilitates interactions between SUR1-TMD0 and Kir6.2. Once assembled, the interactions between SUR1-TMD0 and Kir6.2 become the primary anchor holding the two subunits together and the KNt-SUR1 interaction, rather than serving a structural role, adopts the role of gating regulation (see below).

The location of the KNt density observed in our structures, including the ATP-only structure, also illuminates how Kir6.2 and SUR1 interact to modulate channel activity. First, by occupying the central cavity of the SUR1 ABC core, KNt prevents NBD dimerization and stabilizes SUR1 in an inward-facing conformation, thereby abolishing the ability of Mg-nucleotides to stimulate channel activity (*Sikimic et al., 2019*). Second, by trapping KNt in the central cavity, SUR1 prevents the Kir6.2 tetramer from undergoing conformational changes needed for channel opening. Our results that DTT exposure recovers currents of channels formed by SUR1-C1142 and Kir6.2-L2C previously exposed to I$_2$ indicates crosslinking-induced decrease of channel activity and lends support to the second mechanism. It also explains early studies showing that deletion of KNt increased channel open probability and decreased the ability of sulfonylureas to inhibit the channel (*Babenko and Bryan, 2002*; *Koster et al., 1999a*; *Reimann et al., 1999*; *Shyng et al., 1997*). Interestingly, Wu et al. recently showed that inserting increasing number of amino acids in the linker between SUR1 and Kir6.2 in the SUR1-Kir6.2 fusion channels caused a gradual decrease in channel inhibition by GBC (*Wu et al., 2018*). They postulate that KNt acts as a chain which prevents Kir6.2 cytoplasmic domain from rotating to open the channel when trapped in the SUR1's central cavity; increasing the linker length in the fusion protein reduces the tension exerted by the KNt-SUR1 interaction to reduce channel inhibition by GBC. Future determination of open channel structures will allow direct structural comparison of conformational differences in the Kir6.2 cytoplasmic domain. It is worth noting that the KNt density, albeit weaker, is found in the ATP-only structure. This suggests KNt spends significant time in SUR1's ABC core cavity when Kir6.2 is in the ATP-bound closed state. By contrast, the KNt cryoEM density is absent in the apo structure, suggesting that when no ATP is bound to Kir6.2, KNt has little residence time in the ABC core central cavity. The fact that despite not seeing KNt, the Kir6.2 channel is still closed in the dominant class structure is likely due to loss of endogenous PIP$_2$ during purification.

Our structures show that CBZ and RPG share the same binding pocket as GBC in K$_{ATP}$ channels. Although both RPG and CBZ have been characterized extensively, with RPG being a commonly used oral hypoglycemic medication and CBZ an anticonvulsant, there are no known structures of these drugs bound to proteins. Thus, our structures provide the first examples of how these drugs interact with their target proteins. Despite occupying a common pocket, each of the three compounds appear to form distinct chemical interactions with residues lining the binding pocket. This is particularly clear when comparing RPG with GBC and CBZ. Curiously, we found that the density corresponding to CBZ cannot be fit with a single molecule. CBZ is a well-known Na$_v$ channel blocker (*Lipkind and Fozzard, 2010*). It will be important to determine whether CBZ adopts a similarly bound arrangement in other target proteins.

Despite functional divergence, all ABC transporters share some structural similarities, in particular the ABC core structure (*Thomas and Tampé, 2018*). An exciting finding here is how the Kir6.2

N-terminus can act as a plug or handle to regulate channel gating and assembly by inserting itself into the SUR1 ABC core central cavity formed by the two TM bundles. The exploitation of this structural space is akin to the mechanism by which viral peptide ICP47 enables immune-evasion of the pathogens (*Blees et al., 2017*; *Oldham et al., 2016*; *Parcej and Tampé, 2010*). In this case, ICP47 secreted by viruses such as Herpes simplex virus and Cytomegalovirus inserts into the inner vestibule formed by the ABC peptide transporters TAP-1 and TAP-2 (transporter associated with antigen processing) (*Blees et al., 2017*; *Oldham et al., 2016*), thus preventing the transport of cytosolic peptides into the ER for MHC complex loading and immune surveillance. Following this, molecules which can reside within this space may offer opportunities for functional modulation of ABC transporters such as $K_{ATP}$. Indeed, application of a synthetic Kir6.2 N-terminal peptide (a.a. 2–33) to the cytoplasmic face of $K_{ATP}$ channels in isolated membrane patches has been shown to increase $K_{ATP}$ channel open probability (*Babenko and Bryan, 2002*). This is presumably due to competition of the exogenous peptide with the N-terminus of Kir6.2 for binding to the SUR1 ABC core. With regard to the drug binding pocket, it is worth noting that both GBC and CBZ have been reported to bind or are substrates of other ABC transporters (*Zhou et al., 2008*). For example, *Bessadok et al. (2011)* recently showed that the multidrug resistance transporter P-glycoprotein (ABCB1) recognizes several SUR1 ligands including GBC, albeit with much lower affinity. Moreover, CBZ has been shown to correct the trafficking defect of ΔF508 CFTR (ABCC9), the most prevalent mutation underlying cystic fibrosis (*Carlile et al., 2012*). An intriguing possibility is that these other ABC transporters recognize SUR1 ligands through a similar binding pocket identified here via conserved residues within the SUR1 GBC binding pocket such as R1246 and W1297. It would be interesting to determine in the future whether and how CBZ binding affects ΔF508 CFTR structure to correct its processing defect (*Sampson et al., 2013*).

In summary, our study revealed a mechanism by which a diverse set of compounds modulate the gating and assembly/trafficking of the pancreatic $K_{ATP}$ channel, a critical regulator of glucose-stimulated insulin secretion. Our findings may serve as a drug development platform in $K_{ATP}$ channels and other ABC transporters for new pharmacological chaperones or modulators with improved efficacy and specificity.

# Materials and methods

**Key resources table**

| Reagent type (species) or resource | Designation | Source or reference | Identifiers | Additional information |
|---|---|---|---|---|
| Gene (Cricetus cricetus) | ABCC8 (SUR1) | UniProt database | Q09427 | |
| Gene (Rattus norvegicus) | KCNJ11 (Kir6.2) | UniProt database | P70673 | |
| Recombinant adenovirus (Cricetus cricetus) | FLAG-tagged hamster SUR1 | PMID: 28092267 | | FLAG-epitope inserted at the N-terminus of SUR1 and cloned into AdEasy vector for production of adenovirus |
| Recombinant adenovirus (Rattus norvegicus) | Rat Kir6.2 | PMID: 28092267 | N/A | Constructed using the AdEasy vector for production of adenovirus |
| Recombinant adenovirus | tTA | PMID: 28092267 | N/A | Adenovirus for over-expression of Tetracycline-controlled transactivator (tTA) under CMV promoter used for Tet-Off system |

*Continued on next page*

*Continued*

| Reagent type (species) or resource | Designation | Source or reference | Identifiers | Additional information |
|---|---|---|---|---|
| Recombinant DNA reagent (Cricetus cricetus) | FLAG-tagged ham SUR1 in pECE | PMID: 11226335 | N/A | FLAG-epitope inserted at the N-terminus of SUR1 |
| Recombinant DNA reagent (Rattus norvegicus) | Rat Kir6.2 in pcDNA3 | PMID: 14707124 | N/A | |
| Cell line (Rattus norvegicus) | INS-1 clone 832/13 | PMID: 10868964 | RRID:CVCL_7226 | |
| Cell line (Chlorocebus aethiops) | COSm6 | PMID: 11226335 | RRID:CVCL_8561 | |
| Chemical compound, drug | Digitonin | Calbiochem | CAS 11024-24-1 | |
| Chemical compound, drug | ATP | Sigma-Aldrich | A7699 | |
| Chemical compound, drug | Glibenclamide | Sigma-Aldrich | G0639 | |
| Chemical compound, drug | Repaglinide | Sigma-Aldrich | R9028 | |
| Chemical compound, drug | Carbamazepine | Sigma-Aldrich | C4024 | |
| Chemical compound, drug | CBDPS-H8/D8 | Creative Molecules, Inc | Cat. Number: 014S | |
| Chemical compound, drug | FuGENE6 | Promega | E2691 | |
| Peptide | FLAG-peptide | Sigma-Aldrich | F3290 | |
| Antibody | Anti-FLAG M2 affinity gel | Sigma-Aldrich | A2220 | |
| Antibody | Anti-SUR1 (rabbit polyclonal | PMID: 17575084 | N/A | (1:100) |
| Antibody | Horseradish Peroxidase conjugated goat anti-rabbit secondary | Jackson ImmunoResearch | Code: 111-035-144 | (1/1000) |
| Software, algorithm | Serial EM | PMID: 16182563 | http://bio3d.colorado.edu/SerialEM | |
| Software, algorithm | MOTIONCOR2 | PMID: 28250466 | http://msg.ucsf.edu/em/software/motioncor2 | |
| Software, algorithm | CTFFIND4 | PMID: 26278980 | http://grigorieflab.janelia.org/ctffind4 | |
| Software, algorithm | DoGPicker | PMID: 19374019 | https://sbgrid.org/software/titles/dogpicker | |
| Software, algorithm | Relion-3 | PMID: 30412051 | https://www2.mrc-lmb.cam.ac.uk/relion | |
| Software, algorithm | COOT | PMID: 20383002 | http://www2.mrc-lmb.cam.ac.uk/personal/pemsley/coot | |
| Software, algorithm | UCSF Chimera | PMID: 15046863 | http://www.cgl.ucsf.edu/chimera | |
| Software, algorithm | Pymol | Schrödinger | https://pymol.org/2 | |

## Cell lines used for protein expression

INS-1 cells clone 832/13 and COSm6 cells were used for protein expression (see below). The identity of these cell lines has been authenticated (see Key Resources Table above). These cell lines are not on the list of commonly misidentified cell lines maintained by the International Cell Line

Authentication Committee. The mycoplasma contamination testing was performed routinely in the lab and shown to be negative for the work described here.

## Protein expression and purification

$K_{ATP}$ channels were expressed and purified as described previously (*Martin et al., 2017a*; *Martin et al., 2017b*). Briefly, the genes encoding pancreatic $K_{ATP}$ channel subunits, which comprise a hamster SUR1 and a rat Kir6.2 (94.5% and 96.2% sequence identity to human, respectively), were packaged into recombinant adenoviruses (*Lin et al., 2005*; *Pratt et al., 2009*); both are WT sequences, except for a FLAG tag (DYKDDDDK) engineered into the N-terminus of SUR1 for affinity purification. INS-1 clone 832/13 (*Hohmeier et al., 2000*), a rat insulinoma cell line, was infected with the adenoviral constructs in 15 cm tissue culture plates. Protein was expressed in the presence of 1 mM Na butyrate as well as 5 µM RPG (for the RPG/ATP structure) or 10 µM CBZ (for the CBZ/ATP structure) to enhance expression and formation of the channel complex (*Chen et al., 2013*; *Yan et al., 2006*). At 40–48 hr post-infection, cells were harvested and cell pellets flash frozen in liquid nitrogen and stored at −80˚C until purification.

For purification, cells were resuspended in hypotonic buffer (15 mM KCl, 10 mM HEPES, 0.25 mM DTT, pH 7.5) and lysed by Dounce homogenization. The total membrane fraction was resuspended in buffer A (0.2M NaCl, 0.1M KCl, 0.05M HEPES, 0.25 mM DTT, 4% sucrose, pH 7.5) and solubilized with 0.5% Digitonin. Note for the ATP-only dataset, 1 mM ATP was included throughout the purification; for the CBZ/ATP dataset, 10 µM CBZ and 1 mM ATP were present throughout; for the RPG/ATP dataset, 1 mM ATP and 30 µM RPG were included throughout. The soluble fraction was incubated with anti-FLAG M2 affinity agarose for 4 hr. The agarose beads were washed three times with 4X beads volume of buffer A (without sucrose) and bound proteins eluted with buffer A (without sucrose) containing 0.25 mg/mL FLAG peptide. Purified complexes were concentrated to ~1–1.5 mg/mL and used immediately for cryo grid preparation. For the RPG/ATP sample, the final concentration of RPG was 30 µM, and ATP 1 mM; for the CBZ/ATP sample, the final concentration of CBZ was 10 µM, and ATP 1 mM; for the ATP-only sample, the final ATP concentration was 1 mM.

## Sample preparation and data acquisition for cryo-EM analysis

3 µL of purified $K_{ATP}$ channel complex was loaded onto UltrAufoil gold grids which had been glow-discharged for 60 s at 15 mA with a Pelco EasyGlow. The sample was blotted for 2 s (blot force −4; 100% humidity) and cryo-plunged into liquid ethane cooled by liquid nitrogen using a Vitrobot Mark III (FEI).

Single-particle cryo-EM data was collected on a Titan Krios 300 kV using a Falcon III detector (Thermo Scientific) for the RPG/ATP dataset and a Gatan K2 Summit detector for the CBZ/ATP and the ATP only datasets. The apo dataset was collected on a Talos Arctica 200 kV microscope with a Gatan K2 Summit detector using SerialEM. Data collected using the Gatan K2 Summit direct electron detector were performed in the super-resolution mode, post-GIF (20 eV window), at a physical pixel size of 1.72 Å (Krios) or 1.826 Å (Arctica). For the RPG/ATP dataset, the calibrated pixel-size of the Falcon III was at 1.045 Å. Defocus was varied between −1.0 and −3.0 µm across the datasets. Detailed imaging parameters are provided in *Table 1*.

## Image processing

The raw frame stacks were gain-normalized and then aligned and dose-compensated using Motioncor2 (*Zheng et al., 2017*) with patch-based alignment (5 × 5). CTF parameters were estimated from the aligned frame sums using CTFFIND4 (*Rohou and Grigorieff, 2015*). Particles were picked automatically using DoGPicker (*Voss et al., 2009*) with a broad threshold range to reduce bias. Subsequently, each image was analyzed manually to recover particles missed by automatic picking and remove bad micrographs. 2D classifications were done using RELION-2 (*Kimanius et al., 2016*). Classes displaying fully assembled complexes and high signal/noise were selected and particles re-extracted at 1.72 Å/pix (or 1 Å/pix for the RPG/ATP dataset) and then used as input for 3D classification in RELION-2 (see *Table 1* and *Figure 1—figure supplements 1–4*).

Extensive 3D classification was performed to sample heterogeneity within the data. Symmetry was not imposed at this step in order to select only true four-fold classes. Up to four consecutive rounds of classification were performed, specifying 4 or five classes per round. Individual classes and

combinations of classes were refined for each dataset to achieve the best reconstructions (*Figure 1—figure supplements 1–4*). A soft mask encompassing the entire complex was used during refinement in RELION, with C4 symmetry imposed to yield an overall channel reconstruction using the gold-standard FSC cutoff. The map was B-factor corrected and filtered using RELION-2's Postprocessing procedure, with the same mask used for refinement.

## Focused refinement

Focused refinement of SUR1 was carried out in RELION-3 using symmetry expansion, partial signal subtraction that removes signals outside the masked region, followed by further 3D refinement of signal subtracted particles (*Scheres, 2016*). Different masking strategies were applied to the different datasets for focused refinement to obtain the best reconstruction (*Figure 1—figure supplements 1–4*). For the RPG/ATP dataset, refinement against the SUR1 ABC core module following symmetry expansion and signal subtraction led to significantly improved resolution (*Figure 1—figure supplement 1*). For the CBZ/ATP dataset, refinement against the entire SUR1 was performed, which also yielded an improved density map for SUR1 (*Figure 1—figure supplement 2*).

Similar focused refinement strategies were initially applied for the previously published GBC/ATP dataset and the new ATP-only and the apo datasets. However, this resulted in deterioration of map quality and resolution compared to the starting reconstructions. Therefore, alternative signal subtraction strategies were tested for focused refinement, which led to the use of a mask that includes the Kir6.2 tetramer and one SUR1 for the three datasets. Thus, for the GBC/ATP dataset, the particles included in the final C4 reconstruction were subjected to C4 symmetry expansion and signal subtraction using a mask that includes the Kir6.2 tetramer and one SUR1 (*Figure 1—figure supplement 3*). Although the resulting map showed little improvement in overall resolution compared to our previously published C4 map (*Martin et al., 2017a*), the local resolution of the SUR1 in several regions was significantly improved, especially NBD1 and the linker between TMD2 and NBD2 (*Figure 1—figure supplement 5*). A similar scheme with some modifications was employed for the ATP-only and the apo datasets (*Figure 1—figure supplement 4*).

## Modeling

Modeling was performed for the RPG- and GBC-bound SUR1 maps, which following focused refinement have the highest resolutions among all the structures included in this study.

For RPG-bound SUR1, the final map was B-factor corrected and filtered using RELION-3's Postprocessing tool to optimize observable side chain features. For model building, we used SUR1 from our previously published structure (PDB:6BAA) as the initial model. The ABC core structure including the two transmembrane domains (TMD1 and TMD2) and two cytosolic nucleotide binding domains (NBD1 and NBD2) was docked into the density map by rigid-body fitting using Chimera's 'Fit in' tool. The model was further optimized by rigid-body refinement using 'Phenix.real_space_refinemet' with default parameters (*Afonine et al., 2018*). Of note, additional densities not observed in the previously published $K_{ATP}$ cryoEM density maps are now clearly resolved, particularly the ATP density in NBD1 and the density for the linker regions between TMD2 and NBD2 (*Figure 1—figure supplement 5*). To build additional residues in these regions, the final map was sharpened by B-factor using RELION 3.0 (*Zivanov et al., 2018*) to optimize observable map features. As there are no homology models available for these disordered regions, models were built manually de novo in COOT (*Emsley et al., 2010*) followed by refinement in PHENIX, as detailed below. Initially, poly-alanines were built into the maps with some clear bulky side chains in COOT; refinement was performed by 'Phenix.real_space_refine' with the following parameters: global minimization, morphing, and SUR1 initial model as a restraining reference model. Bulky side chains in the new loop regions were added, and the region between D1060-C1079 that was previously modeled as a non-helical structure was corrected to a complete helix by adjustment of residues in accordance with the density map. The model was refined by iterative manual inspection and side chains adjustment to fit the densities in COOT and real space refinement using 'Phenix.real_space_refine' with tight secondary structure and torsion angle restrains. Previously published NBD domains (PDB: 6BAA) was inserted as a reference model to provide additional restrain and to minimize overfitting. The Kir6.2 N-terminus (amino acids 1–19) was built as a poly-alanine model with the side chain of L2 modeled to show its relation to SUR1 C1142. For modeling RPG in the cryoEM density map, a

molecular topology profile for RPG was created using eLBOW in PHENIX, and refined into the cryo-EM density in COOT. The RPG molecule with new coordination was then added to the ABC core structure model for further refinement.

Model building for the GBC-SUR1 map was similarly performed. Because the GBC-bound SUR1 map was refined using the Kir6.2 tetramer with one full SUR1 subunit as a module, we created an initial model for SUR1 by merging TMD0 from the previous model (PDB:6BAA) with the final ABC core structure built using the RPG-SUR1 map described above, which was then docked into the GBC-SUR1 map using Chimera and refined using COOT and PHENIX. GBC was docked into the ligand density in the GBC-bound SUR1 map derived from focused refinement. In addition to the regions noted above for the RPG-SUR1 map, densities corresponding to K329-G353 are of sufficient quality to build poly-alanines with some bulky side chains de novo in COOT. Extra density at Asn10 likely corresponds to glycosylated Asn10, although the sugar moiety is insufficiently resolved for modeling.

SUR1 model building in the CBZ-bound SUR1 map was also performed following the same procedure outlined for GBC-SUR1, except that the initial model was the final ABC core structure from the RPG-SUR1 model. Note for the CBZ map, no attempt was made to incorporate the CBZ molecule due to uncertainty in whether the CBZ binds as a monomer or dimer, as discussed in the main text. Also, due to insufficient resolution ATP molecule bound at NBD1 was not modeled.

## Functional studies

Point mutations were introduced into hamster SUR1 cDNA in pECE using the QuikChange site-directed mutagenesis kit (Stratagene). Mutations were confirmed by DNA sequencing. Mutant SUR1 cDNA in pECE and rat Kir6.2 in pcDNA3 were co-transfected into COS cells using FuGENE6, as described previously (*Devaraneni et al., 2015*), and used for Western blotting and electrophysiology as described below.

For Western blotting, cells were lysed in a buffer containing 20 mM HEPES (pH 7.0), 5 mM EDTA, 150 mM NaCl, 1% Nonidet P-40, and cOmplete protease inhibitors (Roche) 48–72 hr post-transfection. To rescue trafficking-impaired SUR1-F27S mutant, 0.1% DMSO (vehicle control), 1 μM GBC, 1 μM RPG or 10 μM CBZ were added to cells 24 hr before cell harvest. Proteins in cell lysates were separated by SDS/PAGE (8%), transferred to nitrocellulose membrane, probed with rabbit anti-SUR1 antibodies against a C-terminal peptide of SUR1 (KDSVFASFVRADK), followed by HRP-conjugated anti-rabbit secondary antibodies (Amersham Pharmacia), and visualized by chemiluminescence (Super Signal West Femto; Pierce) with FluorChem E (ProteinSimple).

For electrophysiology experiments testing the effects of crosslinking, cells co-transfected with SUR1 and Kir6.2 cDNAs along with the cDNA for the green fluorescent protein GFP (to facilitate identification of transfected cells) were plated onto glass coverslips 24 hr after transfection and recordings made in the following two days. All experiments were performed at room temperature as previously described *Devaraneni et al. (2015)*. Micropipettes were pulled from non-heparinized Kimble glass (Fisher Scientific) on a horizontal puller (Sutter Instrument, Co., Novato, CA, USA). Electrode resistance was typically 1–2 MΩ when filled with K-INT solution containing 140 mM KCl, 10 mM K-HEPES, 1 mM K-EGTA, pH 7.3. ATP was added as the potassium salt. Inside-out patches of cells bathed in K-INT were voltage-clamped with an Axopatch 1D amplifier (Axon Inc, Foster City, CA). ATP (as the potassium salt), $I_2$ (diluted from 250 mM stock in ethanol), or dithiothreitol (DTT) were added to K-INT as specified in the figure legend. All currents were measured at a membrane potential of −50 mV (pipette voltage = +50 mV). Data were analyzed using pCLAMP10 software (Axon Instrument). Off-line analysis was performed using Microsoft Excel programs. Data were presented as mean ± standard error of the mean (s.e.m.).

## Crosslinking and mass spectrometry

Purified $K_{ATP}$ channels were crosslinked using a 50:50 mixture of light (H8) and heavy (D8), amine-reactive, homobifunctional crosslinker CyanurBiotinDimercaptoPropionylSuccinimide (CBDPS) [14 Å span] (Creative Molecules Inc) (*Petrotchenko et al., 2011*). CBDPS was dissolved in anhydrous DMSO at 100 mM concentration, then immediately added to the purified sample. The final reaction buffer contained 150 mM NaCl, 50 mM KCl, 50 mM HEPES, pH 7.5, 0.05% digitonin, 1 mM CBDPS, 1% DMSO, 1 μM GBC, and 0.30 mg/ml purified protein in a volume of 180 μl. The crosslinking

reaction was allowed to proceed for 20 min on ice, then quenched with 100 mM Tris, pH 8.0. Protein was then methanol/chloroform precipitated by: addition of 400 µl of methanol, vortexing, addition of 100 µl of chloroform, vortexing, addition of 300 µl of water, vortexing, and the mixture centrifuged at 16,000 x g at 4°C for 10 min. The upper aqueous layer was then removed, being careful not to remove the precipitated protein at the interface, 500 µl of methanol was added, the sample vortexed, then centrifuged as above. The supernatant was then removed and the pellet was washed twice by addition of 500 µl of methanol, gentle vortexing, and centrifugation as above. The pellet was dissolved by shaking for 30 min at 37°C in 15 µl of 8M urea and 20 µl of 50 mM ammonium bicarbonate containing 0.2% ProteaseMAX detergent (Promega). The sample was then reduced by addition of 40.3 µl of 50 mM ammonium bicarbonate, 1 µl of 0.5M dithiothreitol, and heating at 57°C for 30 min. The sample was then alkylated by addition of 2.7 µl of 0.55M iodoacetamide and incubation in the dark for 15 min. Digestion of the sample was then performed overnight at 37°C with shaking after addition of 1 µl of 1% ProteaseMAX detergent and 20 µl of MS-Grade trypsin (Thermo Scientific) dissolved at 0.1 µg/µl concentration in 1 mM HCl. Following digestion, trifluoroacetic acid was added to a final 0.5%, the sample incubated at room temperature for 5 min, centrifuged at 16,000 x g for 15 min, and the supernatant removed. Crosslinked peptides were then affinity purified using reagents and hardware provided in a Cleavable ICAT Reagent Kit for Protein Labeling (monoplex version) using the manufacturer's recommended protocol (Sciex), except the digest was applied directly to the avidin affinity cartridge without the prior cation exchange purification, the digest was diluted in 1.0 ml the manufacturer's avidin cartridge loading buffer, and the pH of the solution was adjusted to seven by addition of 1.0 M Tris, pH 8.0 buffer prior to injection onto the avidin cartridge. Purified peptides were then dried by vacuum centrifugation, dissolved in 20 µl of 5% formic acid and analyzed by liquid chromatography/mass spectrometry. The digest was injected onto an Acclaim PepMap 100 µm x 2 cm NanoViper C18, 5 µm trap (Thermo Scientific), at 5 µl/min for 5 min in mobile phase A containing water, 0.1% formic acid, then switched on-line to a PepMap RSLC C18, 2 µm, 75 µm x 25 cm EasySpray column (Thermo Scientific). Peptides were then eluted using a 7.5–30% mobile phase B (acetonitrile, 0.1% formic acid) gradient over 90 min at a 300 nl/min flow rate. Data-dependent tandem mass spectrometry analysis was performed using an Orbitrap Fusion instrument fitted with an EasySpray source (Thermo Fisher Scientific). Survey scans (m/z = 400–1500) and data-dependent MS2 scans were performed in the Orbitrap mass analyzer at a resolution = 120,000, and 30,000, respectively, following higher energy collision dissociation (HCD) using a collision energy of 35 following quadrupole isolation at a 1.6 m/z isolation width. Peptides of charge states 3–7 were selected with signal intensities over $5 \times 104$ and having a targeted inclusion mass difference of 8.05 to select peptides containing the mass shifted CBDPS cross-linkers. The instrument was also configured to collect MS/MS scans for only the heavy labeled peptide pair. The method also used dynamic exclusion with 30 s duration and mass tolerance of 10 ppm. Cross-linked peptides were identified using version 2.0.0.5 of MeroX software (*Iacobucci et al., 2018*) using cross-linker masses of 509.0974 and 517.1476 for the (H8) and (D8) forms of the CBDPS cross-linker respectively, and mass precision tolerances of 5 and 10 ppm for precursors and fragment ions, respectively. The instrument mzML file, detailed instrument settings, MeroX result file, FASTA file containing Kir6.2 and SUR1 sequences, and MeroX settings file can be downloaded at ProteomeXchange Consortium via the PRIDE (*Perez-Riverol et al., 2019*) partner repository with the dataset identifier PXD014498.

## Note added in proof

While this paper was in review, a paper has been published by Ding et al. showing a structure of a $K_{ATP}$ channel formed by a SUR1-Kir6.2 fusion protein bound to repaglinide (*Ding et al., 2019*).

## Acknowledgements

We thank Dr. Christopher B Newgard for the INS-1E cell line clone 832/13 and the staff of the Multi-Scale Microscopy Core at the Oregon Health and Science University for help with imaging and data collection. We also thank Dr. Bruce Patton for comments on the manuscript. This work was supported by US National Institutes of Health grants DK57699 (S-LS), DK066485 (S-LS) and F31 DK105800 (GMM).

## Additional information

### Funding

| Funder | Grant reference number | Author |
|---|---|---|
| National Institutes of Health | DK57699 | Show-Ling Shyng |
| National Institutes of Health | DK066485 | Show-Ling Shyng |
| National Institutes of Health | F31 DK105800 | Gregory M Martin |

The funders had no role in study design, data collection and interpretation, or the decision to submit the work for publication.

### Author contributions

Gregory M Martin, Conceptualization, Data curation, Formal analysis, Funding acquisition, Investigation, Visualization, Methodology, Writing—original draft, Writing—review and editing; Min Woo Sung, Data curation, Formal analysis, Investigation, Visualization, Writing—review and editing; Zhongying Yang, Data curation, Investigation, Visualization; Laura M Innes, Balamurugan Kandasamy, Formal analysis, Investigation; Larry L David, Formal analysis, Investigation, Visualization, Writing—original draft; Craig Yoshioka, Resources, Data curation, Software, Formal analysis, Supervision, Validation, Investigation, Methodology, Writing—original draft; Show-Ling Shyng, Conceptualization, Resources, Data curation, Formal analysis, Supervision, Funding acquisition, Investigation, Visualization, Methodology, Writing—original draft, Project administration, Writing—review and editing

### Author ORCIDs

Craig Yoshioka https://orcid.org/0000-0002-0251-7316
Show-Ling Shyng https://orcid.org/0000-0002-8230-8820

### Decision letter and Author response

Decision letter https://doi.org/10.7554/eLife.46417.040
Author response https://doi.org/10.7554/eLife.46417.041

## Additional files

### Supplementary files

• Transparent reporting form
DOI: https://doi.org/10.7554/eLife.46417.018

### Data availability

CryoEM density maps and PDB files have been desposited in the Worldwide Protein Data Bank (wwPDB) with the following accession codes: SUR1-RPG-ATP state (EMD-20528, PDB ID 6PZ9); SUR1-GBC-ATP state (EMD-20530, PDB ID 6PZA); SUR1-CBZ-ATP state (EMD-20534, PDB ID 6PZC); SUR1-ATP-only state (EMD-20535, PDB ID 6PZI); SUR1-Apo State (EMD-20533, PDB ID 6PZB). Mass spectrometry data has been deposited in the PRIDE database under the project accession ID: PXD014498.

The following datasets were generated:

| Author(s) | Year | Dataset title | Dataset URL | Database and Identifier |
|---|---|---|---|---|
| Shyng SL, Yoshioka C, Martin GM, Sung MW | 2019 | Cryo-EM structure of the pancreatic beta-cell SUR1 bound to ATP and repaglinide | https://www.rcsb.org/structure/6PZ9 | RCSB Protein Data Bank, 6PZ9 |
| Shyng SL, Yoshioka C, Martin GM, Sung MW | 2019 | Cryo-EM structure of the pancreatic beta-cell SUR1 bound to ATP and repaglinide | https://www.emdatare-source.org/EMD-20528 | Electron Microscopy Data Bank, EMD-20528 |
| Shyng SL, Yoshioka | 2019 | Cryo-EM structure of the pancreatic | https://www.rcsb.org/ | RCSB Protein Data |

| | | | | |
|---|---|---|---|---|
| C, Martin GM, Sung MW | | beta-cell SUR1 bound to ATP and glibenclamide | structure/6PZA | Bank, 6PZA |
| Shyng SL, Yoshioka C, Martin GM, Sung MW | 2019 | Cryo-EM structure of the pancreatic beta-cell SUR1 bound to ATP and glibenclamide | https://www.emdatare-source.org/EMD-20530 | Electron Microscopy Data Bank, EMD-20530 |
| Shyng SL, Yoshioka C, Martin GM, Sung MW | 2019 | Cryo-EM structure of the pancreatic beta-cell SUR1 Apo state | https://www.rcsb.org/structure/6PZB | RCSB Protein Data Bank, 6PZB |
| Shyng SL, Yoshioka C, Martin GM, Sung MW | 2019 | Cryo-EM structure of the pancreatic beta-cell SUR1 Apo state | https://www.emdatare-source.org/EMD-20533 | Electron Microscopy Data Bank, EMD-20533 |
| Shyng SL, Yoshioka C, Martin GM, Sung MW | 2019 | Cryo-EM structure of the pancreatic beta-cell SUR1 bound to carbamazepine | https://www.rcsb.org/structure/6PZC | RCSB Protein Data Bank, 6PZC |
| Shyng SL, Yoshioka C, Martin GM, Sung MW | 2019 | Cryo-EM structure of the pancreatic beta-cell SUR1 bound to carbamazepine | https://www.emdatare-source.org/EMD-20534 | Electron Microscopy Data Bank, EMD-20534 |
| Shyng SL, Yoshioka C, Martin GM, Sung MW | 2019 | Cryo-EM structure of the pancreatic beta-cell SUR1 bound to ATP only | https://www.rcsb.org/structure/6PZI | RCSB Protein Data Bank, 6PZI |
| Shyng SL, Yoshioka C, Martin GM, Sung MW | 2019 | Cryo-EM structure of the pancreatic beta-cell SUR1 bound to ATP only | https://www.emdatare-source.org/EMD-20535 | Electron Microscopy Data Bank, EMD-20535 |

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
