## [Decision Letter]

Thank you for submitting your article "Mechanism of pharmacochaperoning in K_ATP_ channels revealed by cryo-EM" for consideration by *eLife*. Your article has been reviewed by four peer reviewers, and the evaluation has been overseen by a Reviewing Editor and Richard Aldrich as the Senior Editor. The following individuals involved in review of your submission have agreed to reveal their identity: Vera Moiseenkova-Bell (Reviewer #1); Colin G. Nichols (Reviewer #3); Jeffrey Agar (Reviewer #4).

The reviewers have discussed the reviews with one another and the Reviewing Editor has drafted this decision to help you prepare a revised submission.

Summary:

The reviewers appreciate the importance of the new cryogenic-EM structural data providing key evidence of the commonality of binding sites for different channel inhibitors and pharmacochaperones. This is also the first observation in cryo-EM of ATP binding to NBD1 of SUR1 in the absence of Mg^2+^, which provides an excellent confirmation of earlier photoaffinity labeling studies.

However, as the paper is focused on the hypothesis that the pharmacochaperone effect occurs through drug stabilization of the interaction between the Kir6.2 N-terminus (KNt) and the SUR ABC core domain, revisions are needed to address weaknesses in the evidence for this hypothesis.

Essential revisions:

1) Please make proper acknowledgment of the prior structural evidence, in the 2018 Wu et al. paper, for binding of the KNt to the SUR core domain. This paper is mentioned, en passant, as revealing a weak and disconnected cryoEM density that 'fueled speculation' that the KNt can be located within the central cavity of the SUR1 core. But this is a bit dismissive. The Wu et al. paper makes a rather strong case for this argument, revealing near contiguous density coming off the Kir6.2 subunit towards the cavity and with only a short gap to the density in the cavity, whereas the current manuscript shows what seems to be a more disembodied density in the cavity, and appears to be at no better resolution than that of Wu et al. Moreover, Wu et al. showed additional correlation in so far as there was no density in the SUR1-Kir6.2 fusion structure (in which the N-terminus is not free to act as an extended peptide).

Please also comment on whether in the current structures there is any continuity between Kir6.2 and the density attributed to the N-terminus of Kir6.2 within the core of SUR1 (as seen by Wu et al.), and what in the new structures makes it convincing that it is the N-terminus. The present functional data are not sufficient to make this conclusion.

2) Address the weaknesses in the new experimental evidence for the binding of the KNt and its role in the pharmacochaperone effect:

A) Potentially the strongest new evidence is the MS identification of cross-linked peptides including the KNt and the ABC core. However, there are major concerns that the XL-MS workflow used could produce false positives, particularly because of the absence of an alkylation step to reduce the chance of disulfide scrambling. These experiments should be repeated using an alkylation step and with tighter mass tolerances, using a well-established or validated XL-MS workflow. (Please see the detailed comments of reviewer #4 on this point, included below.) Complete details of the experimental methods and the XL-MS raw data files should be included.

It should also be acknowledged that the cross-linking is done on purified protein and not on channels in membranes, so it is possible the protein adopts conformations it might not normally do under physiological conditions.

On this question of false positive interactions, note that some of the SUR1-SUR1 crosslinking residues seems very far apart in both propeller and quatrefoil structures. For instance, K915 to K1412 looks to be ~70Å yet the linker is only 15A.

B) The electrophysiological tests of cross-linking of the Kir6.2-L2C mutant aim to provide functional evidence for KNt contacting the ABC core domain in an intact membrane under physiological conditions, however, in the opinion of the reviewers and the Reviewing Editor, these current results do not provide convincing support for the hypothesis. The text states that "co-expression of Kir6.2-L2C with WT SUR1 resulted in channels that displayed an accelerated current decrease in I2 (250µM) that was partially reversible by subsequent exposure to DTT[…]" In fact, both wild type channels and Kir6.2-L2C/SUR1-C1142A channels show a similar extent of inhibition during iodine application. If the difference between Kir6.2_L2C and controls is in the rate of current decrease, there is no quantification to support this claim. It is true that there was no DTT reversal apparent in the controls, but this reversal is only apparent in a subset (Figure 5E) of the Kir6.2-L2C records. Were the experiments that showed more profound reversal only the ones with small currents (like the middle record in Figure 5A)? Was the effect of DTT alone tested on wild-type or mutant channels? As for the drug block experiments, how are the effects of iodine differentiated from the effects of rundown (clearly apparent in the right-hand record in Figure 5A)?

It is unclear how best the authors can address these concerns and shortcomings in the physiological cross-linking experiments. Possibly more experiments along these same lines will make the effects more clear. Alternatively, given the apparent distance between the putatively cross-linked sites in the structure, it may be helpful to investigate the effects of a rapid, non-zero-length crosslinker such as a bis-MTS reagent. Although it would be preferable to include an improved version of these experiments, if this is not possible then these experiments should probably just be omitted from the revised manuscript.

C) Although deletions at the N-terminus of Kir6.2 are said to abolish the pharmacochaperone effect (by interfering with the putative interaction of the KNt with the drug bound within the SUR core), there is a very clear pharmacochaperone effect on the ΔN5 mutation in Figure 4—figure supplement 1C (and data are not shown for the ΔN10 mutation). If indeed the KNt deletion prevents the interaction with the SUR-bound drug, it suggests that the pharmacochaperone effect may be independent of the KNt interaction. The weaker (but still not absent) pharmacochaperone effect with the addition of the F27S mutation (panel B) may simply indicate that the KNt deletion destabilizes the protein and the F27S destabilizes it further, with both effects independent of the drug binding. At a minimum, the results with ΔN10 (with no F27S) should be shown, and this alternative interpretation should be discussed.

3) In Figure 3—figure supplement 1B,C, it is not possible to deduce if the mutations in SUR1 had any effect on drug block because it is very hard to distinguish current rundown from block in the records provided. The difficulty is even greater when the currents are very small as for the mutant channel. How was the% current measured in panel D? There really seems to be no steady state block by the drug (see B,C) and there is no clear, rapid onset of block. The problem with these drugs is that they are largely irreversible and they block slowly at low concentrations. It would be better to test a single drug concentration (preferably 100nM) on each patch. Alternatively, the S1238Y mutation of SUR1 could be exploited for comparisons as sulfonylureas block these mutant channels reversibly and at lower affinity.

4) The value of the Rb+ efflux experiments in Figure 3 is unclear and the conclusions drawn from these experiments may be overstated. The drugs were applied for up to 40 min. Is this long enough to affect surface trafficking of SUR1? Can it be concluded if the data primarily represent the effect of the drug on channel inhibition? The authors argue that differences between mutational effects on the ability of repaglinide and carbamazepine to inhibit flux confirm that the mutations are directly in the drug binding site and do not have an indirect-allosteric effect. At first glance, the effects of mutating these residues in the binding pocket on the ability of the two drugs appear to be similar (Figure 3C,D). Is flux a sensitive enough measurement to make such distinctions? Why were mutations tested using Rb+ efflux, rather than patch clamp? Efflux is both determined by the number of channels in the membrane and by the ability of the drug to block the channel. So they can't state 'mutations reduced the sensitivity of the channel to inhibition' – unless they have shown that changes in surface expression do not occur on the time scale of their experiments.

*Reviewer #1:*

In this manuscript, Dr. Shyng and colleagues presented cryo-EM structures of KATP channels bound to pharmacochaperones glibenclamide, repaglinide, and carbamazepine. All three compounds are clearly resolved in these structures and showed that chemically diverse KATP channel inhibitors interact with the channel through the same binding pocket. In addition, this study provided structural information on how N-terminus of the channel interacts with the transporter and stay in the closed conformation.

Manuscript is very well written, a lot of structural and functional data presented to confirm conclusions drawn from the structural work.

Overall, I do not have cryo-EM specific experimental concerns and consider this manuscript be ready for publication in *eLife*.

*Reviewer #2:*

The authors have provided five cryo-EM structures of the KATP channel complex in association with ATP and either glibenclamide (GBC, a refinement of their previous structure), repaglinide (RPG), carbamazepine (CBZ), or no drug, and in the absence of both drug and ATP. The data show that all 3 drugs bind in the same pocket on SUR1. They also identify some density near the drug binding site in the ABC core of SUR1 that they attribute to the unresolved N-terminus of Kir6.2 (as previously suggested by Wu et al. Protein Sci, 2018). The structural data are complemented by data showing the effect of mutations in the drug binding pocket on the ability of drugs to chaperone SUR1 to the plasma membrane (as indicated by complex glycosylation of SUR1).

In addition, the authors provide data showing the effect of drug binding site mutations on Rb+ efflux (which reflects KATP channel activity in the whole cell) and excised patch recordings of the effects of GBC and RPG on KATP currents in WT and W1297 mutant channels. The location of the N terminus within the SUR1 cavity is explored in two ways: by chemical crosslinking followed by mass spectrometry, and by examining the ability to crosslink residues in SUR1 and Kir6.2 by engineered cysteine pairs.

The cryo-EM structures in complex with RPG and CBZ are novel, and should be published. This is also the first observation in cryo-EM of ATP binding to NBD1 of SUR1 in the absence of Mg^2+^, which provides an excellent confirmation of earlier photoaffinity labeling studies. The surface expression data is also interesting and supports the idea that the drugs chaperone SUR1 to the plasma membrane by binding within the same drug binding pocket as that which is linked to channel inhibition. Surprisingly, since the lab has an excellent reputation for KATP channel electrophysiology, the functional data presented in this paper is weak and it is difficult to draw strong conclusions as to the mutational effects on drug binding or on the cross-linking of the N-terminus of Kir6.2 to SUR1 from the data as presented.

Figure 3—figure supplement 1B,C. It is not possible to deduce if the mutations in SUR1 had any effect on drug block because it is very hard to distinguish current rundown from block in the records provided. The difficulty is even greater when the currents are very small as for the mutant channel. How did they measure the% current in panel D – there really seems to be no steady state block by the drug (see B,C) and there is no clear, rapid onset of block. The problem with these drugs is that they are largely irreversible and they block slowly at low concentrations. It would have been better to have tested a single drug concentration (preferably 100nM) on each patch. Alternatively, the S1238Y mutation of SUR1 could be exploited for comparisons as sulfonylureas block these mutant channels reversibly and at lower affinity.

The authors provide persuasive structural evidence as to which residues line the drug-binding pocket. As such, the value of the Rb+ efflux experiments in Figure 3 is unclear and the conclusions drawn from these experiments may be overstated. The drugs were applied for up to 40 min. Is this long enough to affect surface trafficking of SUR1? Can it be concluded if the data primarily represent the effect of the drug on channel inhibition? The authors argue that differences between mutational effects on the ability of repaglinide and carbamazepine to inhibit flux confirm that the mutations are directly in the drug binding site and do not have an indirect-allosteric effect. At first glance, the effects of mutating these residues in the binding pocket on the ability of the two drugs appear to be similar (Figure 3C,D). Is flux a sensitive enough measurement to make such distinctions? Why were mutations tested using Rb+ efflux, rather than patch clamp? Efflux is both determined by the number of channels in the membrane and by the ability of the drug to block the channel. So they can't state 'mutations reduced the sensitivity of the channel to inhibition' – unless they have shown that changes in surface expression do not occur on the time scale of their experiments.

Figure 5. Most of their functional evidence for the 6.2 N-terminus contacting the ABC core domain comes from this figure. The authors state that "co-expression of Kir6.2-L2C with WT SUR1 resulted in channels that displayed an accelerated current decrease in I2 (250µM) that was partially reversible by subsequent exposure to DTT[…]" In fact, both wild type channels and Kir6.2-L2C/SUR1-C1142A channels show a similar extent of inhibition during I2 application. If the difference between Kir6.2_L2C and controls is in the rate of current decrease, there is no quantification to support this claim. It is true that the authors observe no DTT reversal in their controls, but this reversal is only apparent in a subset (Figure 5E) of their Kir6.2-L2C records. Were the experiments that showed more profound reversal only the ones with small currents (like the middle record in Figure 5A)? Did the authors test the effect of DTT alone on wild-type or mutant channels? As for the drug block experiments, how are the authors differentiating the effects of iodine from the effects of rundown (clearly apparent in the right-hand record in Figure 5A)?

*Reviewer #3:*

This is an interesting paper, in a series with the earlier reports of Martin et al. on KATP channel structure, that provides key evidence of the commonality of binding sites for different channel blockers, and intriguing mechanistic connection between the SUR1 subunit and the Kir6.2 subunit. There are a few points that should be addressed:

1) The paper is generally very well written, but should be carefully checked for minor typos and use of commas. One point of contention: 'evidence' is for or against an argument, 'evidence' does not 'establish' anything scientifically. In a couple of places, 'evidence' should be replaced by 'data' or perhaps 'results'.

2) The authors mention the 2018 Wu et al. paper, en passant, as revealing a weak and disconnected cryoEM density that 'fueled speculation' that the KNt can be located within the central cavity of the SUR1 core. But this is a bit dismissive. The Wu et al. paper makes a rather strong case for this argument, revealing near contiguous density coming off the Kir6.2 subunit towards the cavity and with only a short gap to the density in the cavity, whereas the current manuscript shows what seems to be a more disembodied density in the cavity, and appears to be at no better resolution than that of Wu et al. Moreover, Wu et al. showed additional correlation in so far as there was no density in the SUR1-Kitr6.2 fusion structure (in which the N-terminus is not free to act as an extended peptide). This is not to detract from the authors' conclusions here, but more credit should be given to the previous paper.

3) The major potentially definitive evidence in the current paper for the disposition of KNt is the mass spec data. In this regard, it is common to designate a fragmentation pattern as observed in the peptide fragmentation spectrum from tandem MS on the corresponding peptide sequence. Addition of the fragmentation information would make it easier to interpret the data.

4) There are a couple of assertions of likelihood in the Discussion for which the evidence seems a bit thin. In particular, why is it the case that by trapping KNt in the central cavity, SUR1 is 'likely' to prevent a PIP2-bound conformation in Kir6.2? The PIP2 binding site is not in the distal N terminus.

*Reviewer #4:*

This review concerns only the cross-linking MS (XLMS) results. A summary of my review is that the XLMS workflow is not a "typical" workflow, has at least one fundamental flaw and lacks sensitivity, and the cross-linking results could well be false positive.

XLMS is the most technically demanding and specialized type of biological MS experiment. As a result, expert users tend to rely upon well-established XLMS workflows that minimize false-positive IDs. It appears, however, that the authors have created a novel XLMS workflow (i.e. a workflow predicated upon ETD-dissociation of the cross-linked precursor, followed by MS3 of the resulting fragments, followed by atypical XL-MS data analysis), and combined this with novel methods for sample preparation (i.e. no alkylation to prevent S-S scrambling). While the individual components of their workflow (other than omitting alkylation) are reasonable, have they been proven to work well (in combination) during XL-MS?

This reviewer's recommended "fix" is to employ a start-to-finish XL-MS workflow (e.g. sample preparation, cross-linkers, MS, and DA methods) from a well-established group. The leaders in this field are Albert Heck (https://experiments.springernature.com/articles/10.1038/nmeth.3603) and Lan Huang (https://pubs.acs.org/doi/abs/10.1021/acs.analchem.7b04431). Note that all expert groups employ specialized, fit-for-purpose data analysis and validation methods (e.g. XLinkX or equivalent). If the authors (and editor) choose the risky approach of using the author's current workflow, I include suggestions below for some de-risking.

Five major concerns of the current XL-MS methods include: 1) The authors do not appear to employ a well-established XL-MS workflow (or do not reference one). The potential for false positives (via S-S scrambling) increases, the sensitivity of their analysis suffers, and alternative results (i.e. cross-links that don't fit the proposed model) will not be detected. 2) The cross-linking reaction is not described in sufficient detail to repeat or to critique and a reference to the cross-linking method does not appear to have been included. 3) The omission of the reduction step (i.e. for typical reduction and alkylation) is necessary with this cross-linker and was appropriate. However, the omission of the alkylation step will result in "free" cysteine containing peptides, which will certainly exchange (via thiolate-disulfide interchange) with the DSP S-S bond. This study must be performed with a denaturing alkylation step immediately following cross-linking and quench (prior to digestion). 4) No proper cross-link validation step was included. 5) The engineered disulfides should be detected directly by XL-MS analysis (if ETD cleaves all disulfides, the same workflow used by the authors for other cross-links should work here, provided S-S scrambling is prevented by alkylation).

[Editors’ note: the revised article was subsequently rejected after discussions between the reviewers, but the authors were invited to resubmit after an appeal against the decision. The decision letter after the re-review is shown below.]

Thank you for resubmitting your work entitled "Mechanism of pharmacochaperoning in a mammalian K_ATP_ channel revealed by cryo-EM" for consideration by *eLife*. Your article has been re-reviewed by four peer reviewers, and the evaluation has been overseen by a Reviewing Editor and a Senior Editor. The following individuals involved in review of your submission have agreed to reveal their identity: Vera Y Moiseenkova-Bell (Reviewer #1); Colin G. Nichols (Reviewer #3); Jeffrey Agar (Reviewer #4); Michael Puljung (Reviewer #5).

Our decision has been reached after consultation between the reviewers. Based on these discussions and the individual reviews below, we regret to inform you that your work will not be considered further for publication in *eLife*.

We appreciate the valuable new data in the paper, but our expert mass spectrometry reviewer still has serious concerns that the MS crosslinking results are too prone to false positives, without alkylation to prevent disulfide scrambling and a higher m/z resolution analysis of the crosslinking products. Given this concern combined with the weaknesses already pointed out in the functional disulfide crosslinking work, it seems that the central contention of the paper about the interaction between the KNt and the SUR1 core is not adequately supported by the experimental results.

We understand that you may disagree with these concerns, but we consider them too important to overlook. At this point, we are rejecting the submitted manuscript, but we would welcome a new submission of a manuscript that addressed these concerns, most likely with new experimental results. Alternatively, if you think that we have just got it wrong, our rejection of the paper frees you to submit the paper to other journals.

Reviewers #1 and #3 were satisfied with the revisions.

*Reviewer #4:*

The authors did formulate a general response to XL-MS concerns, and this stated (more or less) that the authors did not share my concerns (although they did provide their raw data, which is great) and therefore did not perform additional experiments. I would not myself consider publishing biologically relevant findings using a new method of analysis that hasn't undergone proper validation experiments, or an MS/MS accuracy cutoff of 1 Da, or not validating any putative cross-links with higher accuracy follow-up scans. However, I will admit to being on the more conservative side in such matters.

My serious concern about false positives arises from free thiols that are part of the protein (which could be "lone thiols" or arise from small amounts of native disulfides that never formed). There is a vast literature that not only predicts that thiolates (i.e. free cys on digested peptides) will attack the disulfide bond of DSP, displacing one of the two peptides, but that can even predict the rate constants of this reaction (which will be near their fastest possible rates at the pH used by these authors, exhibiting half-lives on the millisecond timescale). In my own experience, if you have free thiols (on peptides) in the presence of disulfide bonds (on peptides), there will be complete scrambling of disulfide bonds. Here's the most worrisome part- following attack on the DSP disulfide bond by a Cys residue (from another peptide), the leaving group has Lys-conjugated DSP fragment that terminates in a thiolate. This can then attack another DSP cross-linked peptide and form a completely spurious crosslink (which releases another Lys-conjugated DSS fragment, and the mess continues until all crosslinks are artifacts).

This is a fatal design flaw and is why I suggested they alkylate prior to digestion (which they didn't do). This problem of scrambling is more important than any concern about "overalkylation".

Here is a link to a review describing the exchange rates: https://gmwgroup.harvard.edu/files/gmwgroup/files/369.pdf

And here are a few published descriptions of disulfide scrambling in practice:

https://www.ncbi.nlm.nih.gov/pmc/articles/PMC3082428/

https://www.ncbi.nlm.nih.gov/pubmed/17609855

https://www.ncbi.nlm.nih.gov/pubmed/12441146

https://www.ncbi.nlm.nih.gov/pubmed/10972999

https://www.ncbi.nlm.nih.gov/pmc/articles/PMC2143640/

If we assumed for the sake of argument that there were no problems with scrambling, the 1Da MS/MS cutoff is another fatal flaw (irrespective of other aspects of the XLMS experiment).

The authors also use DSP in a way that it has never been used (I asked them to include a reference to their workflow and they didn't). DSP is usually chemically reduced before MS experiments (and compared to a non-reduced experiment). However, these authors extended Huang's concept of the "MS-cleavable" crosslinker from Collisional activation (CID) acting upon a sulfoxide bond (where it has been validated with crosslinkers), to ETD acting upon disulfides (where it has never been validated with crosslinkers, only intact proteins in still-controversial studies). Prior to using such a workflow to reach biologically significant decisions, there are a number of hoops authors jump through, one being a methods development study where they show a histogram of the length of the ID'd crosslinks (where the 70Å cross-link they ID'd would have been a problem). Here I invited them to prove their workflow on their engineered disulfide bond peptides, which like all other of my requests, they declined.

And the authors don't really detect half-DSP conjugated lys residues- they detect peptides that happen to have modifications that sum to 88 +/- 1 Da (and there are a lot of ways to get to that mass).

They use the insoluble crosslinker, rather than water soluble analogue.

Unlike the studies of established labs, they don't bother using isotope labeled DSP to validate their results. These are commercially available.

Most of these concerns are in my previous review.

*Reviewer #5:*

I find the new version of the Martin et al. manuscript to be greatly improved over the original. The inclusion of new data (particularly in Figure 3) was helpful as was their expanded discussion, particularly in reference to the Wu et al. paper. I am still not completely persuaded by their electrophysiology data in Figure 5 (I2 cross linking), but appreciate the changes they made in the main text regarding this figure, which is a much better description of the data. My remaining concerns are minor and can be addressed as changes to the Discussion.

---

## [Author Response]

Essential revisions:1) Please make proper acknowledgment of the prior structural evidence, in the 2018 Wu et al. paper, for binding of the KNt to the SUR core domain. This paper is mentioned, en passant, as revealing a weak and disconnected cryoEM density that 'fueled speculation' that the KNt can be located within the central cavity of the SUR1 core. But this is a bit dismissive. The Wu et al. paper makes a rather strong case for this argument, revealing near contiguous density coming off the Kir6.2 subunit towards the cavity and with only a short gap to the density in the cavity, whereas the current manuscript shows what seems to be a more disembodied density in the cavity, and appears to be at no better resolution than that of Wu et al. Moreover, Wu et al. showed additional correlation in so far as there was no density in the SUR1-Kir6.2 fusion structure (in which the N-terminus is not free to act as an extended peptide).Please also comment on whether in the current structures there is any continuity between Kir6.2 and the density attributed to the N-terminus of Kir6.2 within the core of SUR1 (as seen by Wu et al.), and what in the new structures makes it convincing that it is the N-terminus. The present functional data are not sufficient to make this conclusion.

We regret to have given the reviewers the impression that we were being dismissive of the study by Wu et al. In response, we have revised the manuscript and provided a detailed account of the evidence described in their paper for the KNt cryoEM density (Results and Discussion sections) to properly acknowledge their findings.

The paper by Wu et al. compared their GBC-bound cryoEM structures using the SUR1-Kir6.2 fusion construct with a 39aa linker, their previous GBC bound structure of channels formed by co-expression of SUR1 and Kir6.2 as separate proteins, and the GBC/ATP bound structures also of channels formed by individual SUR1 and Kir6.2 proteins we published. They noted cryoEM density in SUR1’s central cavity in structures of channels formed by separate SUR1 and Kir6.2 proteins but not by the SUR1-Kir6.2 fusion. More specifically, they used our published GBC/ATP structure to show near contiguous density coming off the Kir6.2 subunit towards the SUR1 cavity. They proposed the density likely corresponds to KNt based on previous structure-function data and argued the lack of density in the fusion cryoEM map is due to the large size of the linker that cannot be accommodated in the central cavity.

We have now included additional cryoEM map figures showing that when our GBC/ATP and RPG/ATP-bound cryoEM maps are filtered to a lower resolution (6Å), contiguous density emanating from the first structured residue in Kir6.2 (R32), following along SUR1-L0, and reaching the ABC core central cavity is seen (revised Figure 4—figure supplement 1). While we agree Wu et al. offered a strong case of argument for the KNt density assignment and our own refined maps showing contiguous density further supports the assignment, there could be alternative structural interpretation given the low resolution of this density. For example, the cryoEM density of TMDs-NBDs linkers as well as the NBD1-TMD2 linker is largely missing in all the published structures and there is a possibility that the proposed Kir6.2 N-terminus density actually corresponds to these missing parts. For this reason, we decided to take the time to design biochemical and functional crosslinking experiments before rushing to publication.

Although the reviewers felt that our functional data was not sufficient (however, see our response below), we believe this data further bolsters the structural data and offer insight into the role of the KNt-SUR1 central cavity interaction in modulating channel activity (see revised Figure 6B). Thus, while the cryoEM structure, crosslinking-mass spectrometry, and functional crosslinking results individually do not provide definitive proof for the KNt density assignment, taken together they offer, in our view, compelling experimental evidence for the structural model proposed by Wu et al. and us.

2) Address the weaknesses in the new experimental evidence for the binding of the KNt and its role in the pharmacochaperone effect:A) Potentially the strongest new evidence is the MS identification of cross-linked peptides including the KNt and the ABC core. However, there are major concerns that the XL-MS workflow used could produce false positives, particularly because of the absence of an alkylation step to reduce the chance of disulfide scrambling. These experiments should be repeated using an alkylation step and with tighter mass tolerances, using a well-established or validated XL-MS workflow. (Please see the detailed comments of reviewer #4 on this point, included below.) Complete details of the experimental methods and the XL-MS raw data files should be included.It should also be acknowledged that the cross-linking is done on purified protein and not on channels in membranes, so it is possible the protein adopts conformations it might not normally do under physiological conditions.

We thank reviewer 4 for his comments and offer the following explanations for our experiments. Although the method the reviewer recommended from the Heck lab is a good one, it requires a completely different informatics workup (XlinkX) and is better suited for the analysis of cross-links in more complex digests. The second paper from Lan Huang is a review of cross-linking, but also highlights the sulfoxide containing cleavable crosslinkers developed in their laboratory that was also used by the Heck lab. While a sulfoxide crosslinker could be used to repeat these experiments, and XlinkX software used to interpret results, we don’t feel that it’s necessary for analyzing crosslinks in a purified protein preparation such as ours. We respectfully disagree with the reviewer’s assessment that because we didn’t follow a “mainstream” method that our results are invalid. We and others have used ETD to break disulfide bonds in native proteins in many previous studies. As the disulfide bond in DSP is essentially identical to the disulfide bonds in peptides (-CH2-S-S-CH2-), there is no reason to believe that ETD wouldn’t also preferentially fragment the disulfide bond in DSP, and that is precisely what we have observed. The fact that we could perform the peptide identifications using a common search engine (Sequest) using MS3 data is a strength, not a liability. In fact, using MS3 based peptide identification in conjunction with a cleavable crosslinker is a common workflow when using a cleavable crosslinker, and this is outlined in Figure 2A of the Huang review.

We also appreciate the reviewer’s concern about disulfide bond scrambling that may occur due to our omission of an alkylation step. This is certainly a concern when analyzing disulfides between cysteine residues in proteins. However, it does not pose a problem when using our crosslinking method with DSP, because it would not lead to misidentification of crosslinks. If a free cysteine carried out a disulfide exchange with a DSP crosslink, the resulting mass of the ETD liberated peptide would not lead to an identification, because it wouldn’t have a DSP remnant of 88 mass units. The worst case scenario would just be that we would have a loss of sensitivity if one of the DSP crosslinks underwent a disulfide exchange reaction with a free cysteine. We chose not to include the iodoacetamide alkylation to avoid over-alkylating the proteins, because the usual second addition of DTT that we normally do to scavenge the excess IAA after alkylation is complete couldn’t be done.

Another one of the reviewer’s concerns was that we didn’t describe the cross-linking reaction in detail. We have now provided more experimental details on the reactions in Materials and methods.

The reviewer is correct that the analysis of crosslinks using mass spectrometry is a demanding experiment. However, we feel that sufficient precautions were made to prevent false identification of the reported crosslinks. This is because: 1) the SEQUEST search of the two liberated peptides coming from a single MS2 event had to simultaneously identify unique peptides coming from either SUR1 or Kir6.2 when using a rat Swiss-Prot protein database containing over 8,000 entries, 2) both identified peptides had to have the +88 mass unit increase localized to a lysine residue due to the expected fragmentation of the disulfide bond within the DSP crosslinker, 3) the lysine identified at the crosslinking site had to exhibit a missed trypsin cleavage, 4) the identified peptides in the SEQUEST search had to have less than a 10 ppm mass error, and 5) the two identified peptides in the crosslink had to have a combined mass that matched the mass of the still crosslinked peptide detected in the MS survey scan. We apologize for the error in the mass tolerance of the precursor ions during the SEQUEST search previously given in the Materials and methods section. This has now been corrected.

A suggestion was made to share the raw files so other groups have the option of validation. The raw files used for the crosslinking experiment has been deposited to the ProteomeXchange Consortium via the PRIDE repository (dataset identifier PXD013873, see Materials and methods under the subheading “DSP crosslinking and mass spectrometry”).

Finally, we have acknowledged in the Results section that the crosslinking is done on purified proteins and not on channels in membranes, so it is possible the protein adopts conformations it might not normally do under physiological conditions.

On this question of false positive interactions, note that some of the SUR1-SUR1 crosslinking residues seems very far apart in both propeller and quatrefoil structures. For instance, K915 to K1412 looks to be ~70Å yet the linker is only 15A.

We thank the reviewer for pointing this out. We have now added in the Materials and methods and figure legend that because we use detergent solubilized, purified channel proteins for crosslinking, we cannot rule out the possibility of inter-SUR1 crosslinks. When we interpret the cross-linking/mass spectrometry results we do keep this in mind. The SUR1 K915-K1412 crosslink is very likely a result of inter-SUR1 occurrence within the same channel. As noted by the reviewers, K915 and K1412 are very far apart in the single SUR1 protein in our structure. However, the residues from adjacent SUR1 subunits in the same channel could reach crosslinking distance if SUR1 subunits are flexible. In this regard, it is worth noting that we indeed observe flexibility of SUR1 subunits that deviate from C4 symmetry (Results section). While we cannot rule out inter-channel SUR1 crosslinks, the probability of detecting such random crosslinks would be expected to be quite low. An inter-channel crosslink is even less likely the case for the Kir6.2-K5 and SUR1-K602 pair considering that the Kir6.2 tetramer is surrounded by SUR1 subunits. Nevertheless, we cannot completely rule out the possibility that this crosslink arises from dissociated subunits.

For the caveats discussed above we only used the CL/MS data as supporting evidence in our manuscript (see response to point 1 above). We note however, in our published study examining the effects of GBC and CBZ on photo-crosslinking of azidophenylalanine engineered at amino acid 12 or 18 position of Kir6.2 to SUR1, crosslinking was performed in intact cells. While we were unable to identify SUR1 residues to which Kir6.2-12AzF or 18AzF form crosslinks, our observation that crosslinking is enhanced by GBC and CBZ again provide supporting evidence of pharmacochaperone-dependent KNt interactions with SUR1.

B) The electrophysiological tests of cross-linking of the Kir6.2-L2C mutant aim to provide functional evidence for KNt contacting the ABC core domain in an intact membrane under physiological conditions, however, in the opinion of the reviewers and the Reviewing Editor, these current results do not provide convincing support for the hypothesis. The text states that "co-expression of Kir6.2-L2C with WT SUR1 resulted in channels that displayed an accelerated current decrease in I2 (250µM) that was partially reversible by subsequent exposure to DTT […]" In fact, both wild type channels and Kir6.2-L2C/SUR1-C1142A channels show a similar extent of inhibition during iodine application. If the difference between Kir6.2_L2C and controls is in the rate of current decrease, there is no quantification to support this claim. It is true that there was no DTT reversal apparent in the controls, but this reversal is only apparent in a subset (Figure 5E) of the Kir6.2-L2C records. Were the experiments that showed more profound reversal only the ones with small currents (like the middle record in Figure 5A)? Was the effect of DTT alone tested on wild-type or mutant channels? As for the drug block experiments, how are the effects of iodine differentiated from the effects of rundown (clearly apparent in the right-hand record in Figure 5A)?It is unclear how best the authors can address these concerns and shortcomings in the physiological cross-linking experiments. Possibly more experiments along these same lines will make the effects more clear. Alternatively, given the apparent distance between the putatively cross-linked sites in the structure, it may be helpful to investigate the effects of a rapid, non-zero-length crosslinker such as a bis-MTS reagent. Although it would be preferable to include an improved version of these experiments, if this is not possible then these experiments should probably just be omitted from the revised manuscript.

We appreciate the reviewers’ comments. These experiments turned out to be more challenging compared to our previous crosslinking pairs in the cytoplasmic domain of the channel (including Kir6.2-E229C/Kir6.2-R314C and Kir6.2-Q52C/SUR1-E203C). Cytoplasmic cysteine pairs are easily crosslinked by H_2_O_2_ or copper phenanthroline or even form disulfide bond spontaneously upon patch excision (Lin et al., 2003; Pratt et al., 2012).

For the Kir6.2-L2C and SUR1-C1142 crosslinking, we had to test many different conditions. Initially we tried H_2_O_2_ and copper phenanthroline at different concentrations as we did in prior studies and tested more than 20 SUR1-Kir6.2 cysteine pair combinations, however, no clear reversal of currents by DTT after exposure to oxidizing agents was ever observed. After reviewing membrane protein crosslinking literature, we learned that I2 works better for residues in the transmembrane regions that are less solvent exposed. Indeed, systematic testing of multiple SUR1/Kir6.2 pairs using a paradigm of I2 exposure followed by DTT exposure, we were finally able to see a clear DTT reversal effect on the currents of the SUR1-C1142/Kir6.2-L2C pair. As noted by the reviewers, not every patch showed clear reversal and those that did not tended to show less current reduction with I2 exposure. While we cannot offer definitive explanations, a likely reason is that I2 did not induce significant crosslinking in these patches. This could happen if the patches were deeply recessed and slow to respond to reagents present in the bath solution (this is not uncommon in inside-out recordings based on our >20 years of experience). The other possibility is that channels in those patches had higher open probability such that the probability of KNt reaching the SUR1 central cavity was low. Consistent with the latter explanation, when channels were exposed to KINT solution containing 1mM EDTA which significantly reduces rundown caused by Mg^2+^-dependent loss of PIP2 (Lin et al., 2013), we did not see rapid current decay in I2 nor subsequent reversal by DTT. This is the reason why we performed the crosslinking experiments in KINT without EDTA. Under this condition, WT channels display a variable degree of “rundown” as well appreciated by researchers in the field. To our knowledge what accounts for this variability is not well understood, which makes it difficult to differentiate rundown from I2-induced decrease of currents. In light of the reviewers’ comments we have removed the phrase “accelerated current decrease” and the revised statement now reads: "co-expression of Kir6.2-L2C with WT SUR1 resulted in channels that displayed current decrease in I2 (250μM) that was partially reversible by subsequent exposure to DTT […]".

We agree with the reviewers that performing more experiments will not resolve the intrinsic difficulties of these crosslinking experiments. However, we respectfully disagree that the data presented should be removed as we believe that the DTT-induced current reversal (despite not in every patch) provides strong functional evidence for the structural interpretation and for a role of this interaction in channel activity. Taken as a whole, our structural, biochemical, and functional results converge and corroborate to support the mechanism we propose on how pharmacochaperones inhibit channel activity and correct channel trafficking defects. Nonetheless, we appreciate the weaknesses pointed out by the reviewers and have tried our best to revise the manuscript to explain the crosslinking experiments more clearly (Figure 5 legend).

Finally, regarding testing the effect of DTT alone on WT or mutants, we have documented such control experiments in our previous publications (Lin et al., 2003; Pratt et al., 2012). As such, we do not feel the need to show these recordings again. We have now included in the Figure 5 legend to specifically state that DTT alone does not cause significant channel run-up or run-down aside from an acute reversible blocking effect as indicated by the

arrow in the first recording (Figure 5A) and cited our published papers where this was documented.

C) Although deletions at the N-terminus of Kir6.2 are said to abolish the pharmacochaperone effect (by interfering with the putative interaction of the KNt with the drug bound within the SUR core), there is a very clear pharmacochaperone effect on the ΔN5 mutation in Figure 4—figure supplement 1C (and data are not shown for the ΔN10 mutation). If indeed the KNt deletion prevents the interaction with the SUR-bound drug, it suggests that the pharmacochaperone effect may be independent of the KNt interaction. The weaker (but still not absent) pharmacochaperone effect with the addition of the F27S mutation (panel B) may simply indicate that the KNt deletion destabilizes the protein and the F27S destabilizes it further, with both effects independent of the drug binding. At a minimum, the results with ΔN10 (with no F27S) should be shown, and this alternative interpretation should be discussed.

As requested we have now included a blot of WT-SUR1 co-expressed with ΔN10-Kir6.2 and treated with vehicle, glibenclamide, repaglinide, or carbamazepine (see revised Figure 4—figure supplement 2C) showing a further attenuation of drug effects on the maturation of WT-SUR1, similar to that seen for F27S-SUR1.

Regarding the alternative interpretation that KNt deletions make SUR1 less stable and F27S makes the protein even less stable, our response is as follows. By looking at the SUR1 upper band intensity, this may seem like a logical alternative interpretation. However, an important reminder here is that the upper band only appears when the channel complex is fully formed and has passed the ER quality control, which requires successful assembly of SUR1 with Kir6.2. Thus the reduced upper band indicates less stable SUR1/Kir6.2 complex and not SUR1 alone. We think it is unlikely that KNt deletion significantly impacts SUR1 stability as this would be expected to lead to increased ER associated degradation and decreased lower band intensity, which is not what we observed (see revised Figure 4—figure supplement 2B, C)

How do we explain the residual drug effects seen in SUR1 co-expressed with ΔN5- or even ΔN10-Kir6.2? We believe the most logical interpretation is that the distal ~20 amino acids of KNt is critical for SUR1 and Kir6.2 interactions during the initial assembly process even in the absence of drugs. Drug binding boosts the interaction in part by participation of KNt in drug binding but also by stabilizing SUR1 in an inward-facing conformation that allows KNt to interact with the central cavity for channel assembly. In addition, we have previously shown using metabolic pulse-chase experiments that drug binding slows ER degradation of mutant SUR1 (A116P; Yan et al., 2004), which would allow more time for Kir6.2 to interact before the mutant SUR1 is targeted for ERAD. Progressive deletion of KNt leads to a gradual loss of PCs’ ability to chaperone channel assembly by making KNt an increasingly weaker anchor. We have now revised the Results and Discussion sections and the cartoon figure (Figure 6) to explain our data more clearly.

Reviewer 2 also commented that in our study deletion of Kir6.2 N-terminus markedly reduced maturation of WT SUR1; however, previous studies co-expressing WT SUR1 and Kir6.2Δ5 or Kir6.2Δ10 showed descent currents. We note that using oocyte or mammalian cell transient expression, it is always possible to obtain membrane patches containing sufficient channels from individual cells that do not reflect the overall expression level. Also, because channels lacking the Kir6.2 N-terminus have higher *Po*, it can give the impression that their expression is not affected.

3) In Figure 3—figure supplement 1B,C, it is not possible to deduce if the mutations in SUR1 had any effect on drug block because it is very hard to distinguish current rundown from block in the records provided. The difficulty is even greater when the currents are very small as for the mutant channel. How was the% current measured in panel D? There really seems to be no steady state block by the drug (see B,C) and there is no clear, rapid onset of block. The problem with these drugs is that they are largely irreversible and they block slowly at low concentrations. It would be better to test a single drug concentration (preferably 100nM) on each patch. Alternatively, the S1238Y mutation of SUR1 could be exploited for comparisons as sulfonylureas block these mutant channels reversibly and at lower affinity.

The W1297 functional testing experiments were meant to provide further support to our drug binding site structural model. However, in revising the manuscript, we believe this is not necessary and may cause more confusion (based on reviewers’ feedback). As such, we have removed this supplementary figure.

For reviewers’ information, we have used a similar experimental scheme (10 and 100nM GBC) to study other binding pocket mutations in our recent publication where a number of mutations but not all render GBC inhibition highly reversible (see Figure 7 in Martin et al., eLife, 2017b).

4) The value of the Rb+ efflux experiments in Figure 3 is unclear and the conclusions drawn from these experiments may be overstated. The drugs were applied for up to 40 min. Is this long enough to affect surface trafficking of SUR1? Can it can be concluded if the data primarily represent the effect of the drug on channel inhibition? The authors argue that differences between mutational effects on the ability of repaglinide and carbamazepine to inhibit flux confirm that the mutations are directly in the drug binding site and do not have an indirect-allosteric effect. At first glance, the effects of mutating these residues in the binding pocket on the ability of the two drugs appear to be similar (Figure 3C,D). Is flux a sensitive enough measurement to make such distinctions? Why were mutations tested using Rb+ efflux, rather than patch clamp? Efflux is both determined by the number of channels in the membrane and by the ability of the drug to block the channel. So they can't state 'mutations reduced the sensitivity of the channel to inhibition' – unless they have shown that changes in surface expression do not occur on the time scale of their experiments.

Although the experiments were conducted over a 40 min period, the difference in efflux rate was already apparent during the first few minutes when we look at the efflux profile over time (see Figure 7A in Martin et al., eLife, 2017b). We don’t have any experimental evidence that the number of channels does not change over the course of the efflux experiment in the present study for every mutant. However, our previous surface biotinylation studies have not found significant changes in surface expression of WT channels in COS cells over 30min (e.g. Bruederle et al. Traffic, 2011).

In our previous publication where we tested the effect of GBC binding site mutations on the ability of GBC to inhibit channel activity using Rb efflux assays, we did include an Rb efflux profile example in addition to summary data. In the current study we repeated these experiments using the same set of mutations but testing response to RPG and CBZ with the intent to further validate our structural models for the RPG and CBZ binding pockets. We therefore only presented summary data and referred to our previous publication for the initial GBC experimental dataset. Rb efflux is an efficient way to assess drug response. In our previous study we used both Rb efflux and electrophysiology data to demonstrate effects of binding site mutations (see Figure 7 and Figure 7—figure supplement 1 in Martin et al., eLife, 2017b). As we believed the main point of this experiment is to show reduced inhibition by the mutants to confirm the binding pocket model, we did not think it is necessary to perform electrophysiology experiments especially considering the focus of this paper is on the pharmacochaperoning effects of the drugs. In place of electrophysiology, we performed pharmacochaperone rescue experiments to show that binding pocket mutations also compromised PC rescue effects.

In light of the reviewers’ comments, we have removed the Rb efflux results in Figure 3 and only show the impact of the binding site mutations on chaperoning effects of the drugs. Note, we also replaced the original Western blot showing the effect of RPG with a new one showing the effect of GBC and RPG side-by-side to facilitate direct comparison (now Figure 3C).

[Editors’ note: the author responses to the re-review follow.]

We appreciate the valuable new data in the paper, but our expert mass spectrometry reviewer still has serious concerns that the MS crosslinking results are too prone to false positives, without alkylation to prevent disulfide scrambling and a higher m/z resolution analysis of the crosslinking products. Given this concern combined with the weaknesses already pointed out in the functional disulfide crosslinking work, it seems that the central contention of the paper about the interaction between the KNt and the SUR1 core is not adequately supported by the experimental results.We understand that you may disagree with these concerns, but we consider them too important to overlook. At this point, we are rejecting the submitted manuscript, but we would welcome a new submission of a manuscript that addressed these concerns, most likely with new experimental results. Alternatively, if you think that we have just got it wrong, our rejection of the paper frees you to submit the paper to other journals.Reviewers #1 and #3 were satisfied with the revisions.Reviewer #4:[…] The authors also use DSP in a way that it has never been used (I asked them to include a reference to their workflow and they didn't). DSP is usually chemically reduced before MS experiments (and compared to a non-reduced experiment). However, these authors extended Huang's concept of the "MS-cleavable" crosslinker from Collisional activation (CID) acting upon a sulfoxide bond (where it has been validated with crosslinkers), to ETD acting upon disulfides (where it has never been validated with crosslinkers, only intact proteins in still-controversial studies). Prior to using such a workflow to reach biologically significant decisions, there are a number of hoops authors jump through, one being a methods development study where they show a histogram of the length of the ID'd crosslinks (where the 70 Å cross-link they ID'd would have been a problem). Here I invited them to prove their workflow on their engineered disulfide bond peptides, which like all other of my requests, they declined.And the authors don't really detect half-DSP conjugated lys residues- they detect peptides that happen to have modifications that sum to 88 +/- 1 Da (and there are a lot of ways to get to that mass).They use the insoluble crosslinker, rather than water soluble analogue.Unlike the studies of established labs, they don't bother using isotope labeled DSP to validate their results. These are commercially available.Most of these concerns are in my previous review.

We thank Dr. Agar for his constructive criticisms and valuable suggestions to improve our XL-MS experiment. Two fatal design flaws were pointed out, one concerns potential false positives due to disulfide scrambling, and the other is the 1Da MS/MS cut-off. On the second point, we apologize for not making this clear in our R1 version of the manuscript. The 1.0 Da value is for the MS3 scan in the Orbitrap Fusion’s ion trap and is within the setting commonly used (this low mass accuracy reflects the ion trap's inability to resolve isotopes for sparse or highly charged peptides; please see https://www.ncbi.nlm.nih.gov/pmc/articles/PMC3226415/; https://www.ncbi.nlm.nih.gov/pmc/articles/PMC4966894/), while the precursor tolerance is set at 10 ppm.

Nonetheless, given the reviewer’s serious concerns about our results and that many validation steps were requested to support our data which will take significant amount of time, we decided to repeat the XL-MS experiment using a different crosslinker that has been well documented in a number of published studies.

Specifically, we used an isotopically coded, CID-cleavable, biotinylated amine-reactive homobifunctional crosslinker, CBDPS (14Å space arm) (https://www.ncbi.nlm.nih.gov/pmc/articles/PMC3033670/) for our experiments. The purified proteins were reduced with DTT and alkylated immediately after crosslinking and before trypsin digestion. The digested peptides containing the crosslinker were affinity purified via the biotin tag and MS analysis was conducted with mass precision tolerances of 5 and 10 ppm for precursors and fragment ions, respectively (details in the revised Materials and methods section).

Below are some examples of published studies which have used CBDPS with similar workflow to derive protein structural information:

https://www.ncbi.nlm.nih.gov/pmc/articles/PMC3814369/

https://www.ncbi.nlm.nih.gov/pmc/articles/PMC3642325/

https://www.ncbi.nlm.nih.gov/pmc/articles/PMC5501500/

https://www.ncbi.nlm.nih.gov/pmc/articles/PMC6561701/

https://www.ncbi.nlm.nih.gov/pmc/articles/PMC6453469/

The new experiment identified 2 inter-protein peptides between Kir6.2 and SUR1 and 4 intra-protein peptides in SUR1 as listed below (also shown in the revised Figure 4—figure supplement 3), with the distance between α carbons of the two cross-linked lysine residues in our model indicated where possible.

Kir6.2-SUR1:

Kir6.2 K5-SUR1 K205: 40.7Å? (see discussion below)

Kir6.2 K5-SUR1 K602 (identified in the previous DSP XL-MS experiments): 17.6Å

SUR1-SUR1:

K276-K394: 21Å

K276-K644: K644 is not modeled in the structure but is predicted to be located below K276

K941-K948: in NBD1-TMD2 linker not modeled in the structure but should be close given residue numbers

K1344-K1412: 17.6Å

The crosslinks identified using CBDPS are not identical to those identified in our previous experiments using DSP (12Å space arm, without DTT reduction and alkylation to prevent potential disulfide scrambling). Importantly, however, the same crosslinked peptide linking Kir6.2-K5 and SUR1-K602 was identified in both experiments (please see the revised Figure 4—figure supplement 3). The new XLMS results validate the physical proximity between Kir6.2-K5 and SUR1-K602 in our purified channels under the stated experimental conditions and strongly support our assignment of the Kir6.2 distal N-terminus density in SUR1’s ABC core central cavity. Note, in addition to the crosslink between Kir6.2-K5 and SUR1-K602, we also identified another crosslink between Kir6.2-K5 and SUR1-K205. According to our dominant class of GBC-bound cryoEM structure the Kir6.2 N-terminus is modelled within the SUR1 ABC core central cavity such that the distance between Kir6.2-K5 and SUR1-K205, which is in the proximal L0 region close to the Kir6.2 tetramer would be quite far (40.7Å). However, we should point out that in our cryoEM map, Kir6.2 N-terminus density is relatively poorly resolved compared to the rest of the structure, an indication of its flexibility even in the GBC-bound state. Thus, the identification of this alternative crosslink is consistent with our overall cryoEM structure data and suggests Kir6.2 N-terminus could dissociate from the SUR1 central cavity such that Kir6.2-K5 is within reach of SUR1-K205 via the CBDPS crosslinker.

Because the potential problems raised by the reviewer on our previous DSP crosslink experiments, we have deleted the original figure and methods and replaced them with the new CBDPS XLMS result figure and methods. The raw data have been deposited into the proteomic database PRIDE.